# Temporal chunking as a mechanism for unsupervised learning of task-sets

Flora Bouchacourt[1,2], Stefano Palminteri[1,2,3], Etienne Koechlin[1,2], Srdjan Ostojic[1,2,3]*

[1]Laboratoire de Neurosciences Cognitives et Computationnelles, Institut National de la Sante et de la Recherche Medicale, Paris, France; [2]Departement d'Etudes Cognitives, Ecole Normale Superieure, Paris, France; [3]Institut d'Etudes de la Cognition, Universite de Recherche Paris Sciences et Lettres, Paris, France

**Abstract** Depending on environmental demands, humans can learn and exploit multiple concurrent sets of stimulus-response associations. Mechanisms underlying the learning of such task-sets remain unknown. Here we investigate the hypothesis that task-set learning relies on unsupervised chunking of stimulus-response associations that occur in temporal proximity. We examine behavioral and neural data from a task-set learning experiment using a network model. We first show that task-set learning can be achieved provided the timescale of chunking is slower than the timescale of stimulus-response learning. Fitting the model to behavioral data on a subject-by-subject basis confirmed this expectation and led to specific predictions linking chunking and task-set retrieval that were borne out by behavioral performance and reaction times. Comparing the model activity with BOLD signal allowed us to identify neural correlates of task-set retrieval in a functional network involving ventral and dorsal prefrontal cortex, with the dorsal system preferentially engaged when retrievals are used to improve performance.

## Introduction

When engaged in a given task, humans are capable of learning and exploiting multiple concurrent strategies depending on environmental demands. For instance, in the classical Stroop task (*Stroop, 1935*; *MacLeod, 1991*), an identical stimulus like a colored word leads to different responses depending on whether the current requirement is to read the word or identify its color. Human subjects are able to learn to flexibly switch between these two different stimulus-response associations, often called task-sets (*Sakai, 2008*). Studies of task-set learning predominantly rely on models that describe behavioral learning without direct biological constraints (*Daw et al., 2005*; *Dayan and Daw, 2008*; *Botvinick et al., 2009*; *Daw et al., 2011*; *Alexander and Brown, 2011*; *Russek et al., 2017*; *Franklin and Frank, 2018*). While these models are able to capture computational aspects of behavior, and correlate them with physiological measurements (*Koechlin and Hyafil, 2007*; *Niv, 2009*; *Daw et al., 2011*; *Wilson et al., 2014*; *Alexander and Brown, 2011*), understanding the underlying biologically-inspired mechanisms is an open issue, which requires an intermediate level of description that bridges between biology and behavior (*Wang, 2002*; *Rougier et al., 2005*; *Soltani and Wang, 2006*; *Wong and Wang, 2006*; *Fusi et al., 2007*; *Frank and Badre, 2012*; *Collins and Frank, 2013*; *Bathellier et al., 2013*; *Kuchibhotla et al., 2019*).

One hypothesis (*Rigotti et al., 2010b*) states that learning task-sets, and more generally rule-based behavior, relies on unsupervised learning of temporal contiguity between events. Events that occur repeatedly after each other are automatically associated as demonstrated in classical conditioning experiments (*Rescorla and Wagner, 1972*; *Hawkins et al., 1983*; *Rescorla, 1988*; *Sakai and Miyashita, 1991*; *Kahana, 1996*; *Yakovlev et al., 1998*). If one thinks of individual stimulus-

*For correspondence: srdjan.ostojic@ens.fr

Competing interests: The authors declare that no competing interests exist.

response associations as abstracted events, temporally chunking two or more such events effectively corresponds to learning a simple task-set or association rule. Hebbian synaptic plasticity (*Hebb, 1949*) can naturally lead to such unsupervised learning of temporal contiguity between events (*Földiák, 1991*; *Wallis et al., 1993*; *Griniasty et al., 1993*; *Fusi, 2002*; *Soltani and Wang, 2006*; *Blumenfeld et al., 2006*; *Fusi et al., 2007*; *Preminger et al., 2009*; *Li and DiCarlo, 2010*; *Ostojic and Fusi, 2013*), and therefore provides a simple biological mechanism for learning task-sets (*Rigotti et al., 2010b*) and more generally model-based planning (*Gershman et al., 2012*; *Russek et al., 2017*).

Here we use an abstracted network model to examine the hypothesis that task-sets are learnt through simple Hebbian plasticity based on temporal contiguity between different stimulus-response associations. We test this model on a specific experimental task where human subjects had to acquire multiple sets of stimulus-action pairs (*Collins and Koechlin, 2012*). We first show that the model is able to learn correct task-sets if the plasticity leading to chunking between stimulus-action pairs is slower than the learning of individual stimulus-action associations. Fitting the network model to behavior on a subject-by-subject basis then allowed us to examine specific predictions based on the hypothesis that task-set learning relies on temporal chunking of events. One prediction pertains to the case when a task-set is retrieved correctly, and a second one to the case when this retrieval is maladaptive. We show that both predictions are borne-out by the behavioral data at the level of individual subjects. Moreover, we show that the time-series of the inference signal predicting task-set retrieval in the model correlates with blood oxygen level dependent (BOLD) signal recorded from fMRI (*Donoso et al., 2014*) in a functional network engaging medial and dorsal prefrontal cortex. Altogether, our results demonstrate that a simple mechanism based on Hebbian association of temporally contiguous events is able to parsimoniously explain the learning of complex, rule-based behavior.

## Results

### Behavioral evidence for task-set-driven behavior

To investigate the mechanisms for task-set learning, we examined a specific experiment performed by 22 human subjects (Experiment 1 [*Collins and Koechlin, 2012*], see Materials and methods). In each trial, the subjects had to associate a visual stimulus with a motor response (*Figure 1a*). The subjects needed to learn the correct associations based on a feedback signal, which was misleading in 10% of the trials. The set of correct stimulus-response associations, which we will denote as *task-set* in the following, was fixed during a block of trials of random length (called an *episode*), and changed repeatedly to a non-overlapping set without explicit indication. As the feedback was not fully reliable, the subjects could not directly infer the task-set changes from the feedback on a single trial, but needed to integrate information.

The subjects' behavior was compared between an *open-ended session*, in which the valid task-set was different in each episode, and a *recurrent session* in which only three task-sets appeared repeatedly. In the open-ended session, as each task-set was seen only once, a correct response to one stimulus bore only minimal information about the correct responses to the other stimuli (the responses to the three stimuli had to be different). In contrast, in the recurrent session, a correct response to a given stimulus fully predicted the correct responses to the other stimuli. Learning full task-sets rather than individual associations therefore allowed subjects to increase their performance.

Behavioral data indicated that beyond individual stimulus-response associations, subjects indeed learned task-sets in the recurrent session (*Collins and Koechlin, 2012*). Additional evidence in that direction is displayed in *Figure 1b*, where we show the proportion of correct responses to stimuli seen for the first time after the first correct response in an episode. This quantity was determined for the last third of the session, when the subjects had already experienced several times the three re-occurring task-sets within the recurrent session. If the subjects had perfectly learned the task-sets, they could directly infer the correct response to the newly seen stimulus from a single correct response to another stimulus. The data shows that indeed some subjects perfectly learned full task-sets, so that their performance was maximal after the first correct trial. The performance averaged over all subjects was significantly higher in the recurrent session compared to the open-ended

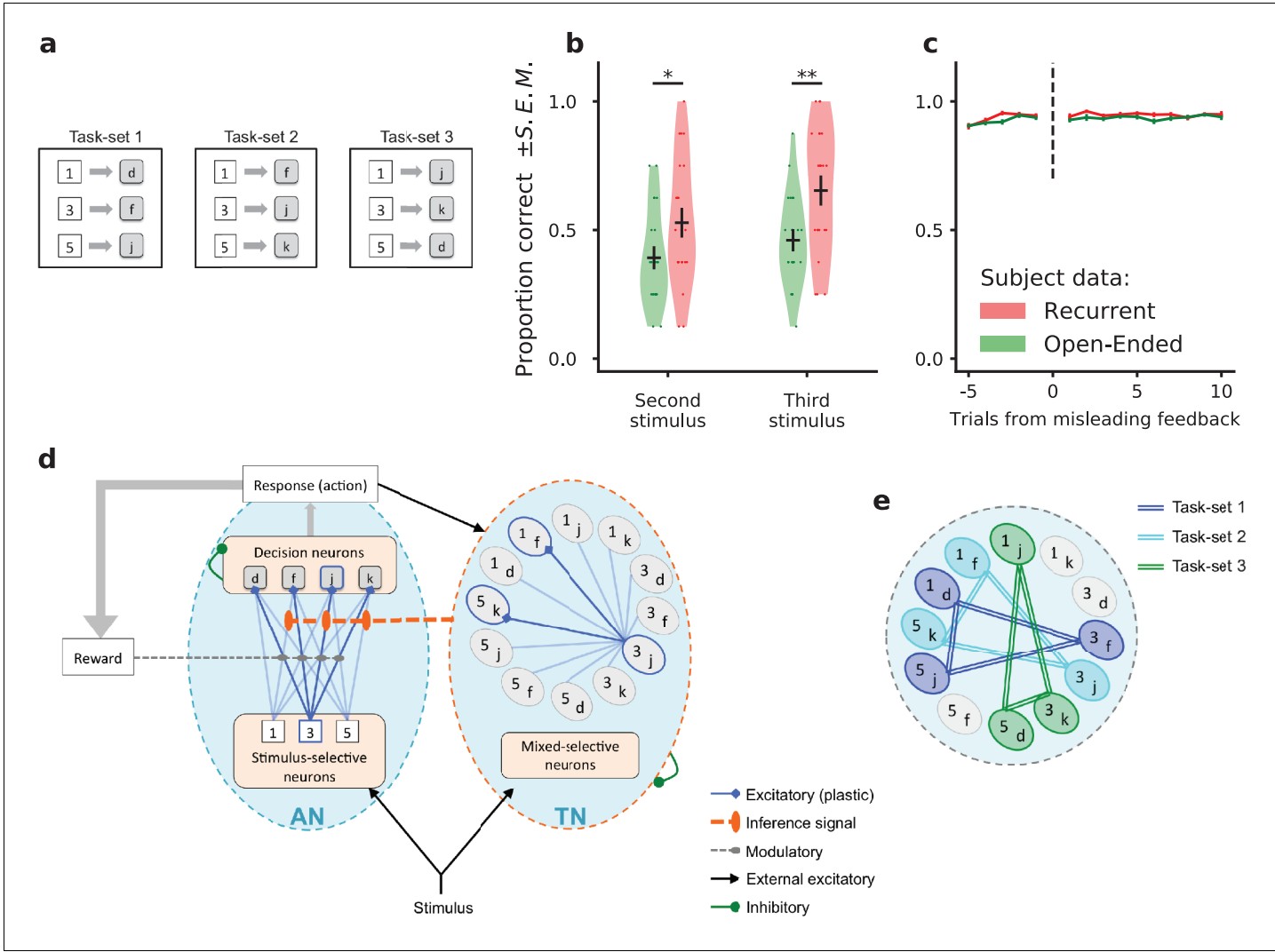

**Figure 1.** Task-set learning experiment and subject behavior. (a) Schematic of the behavioral task. Subjects had to learn associations between visual stimuli (represented here as {1, 3, 5}) and motor responses (represented here as {d, f, j, k}). The set of correct stimulus-response associations, denoted as *task-set*, was fixed during a block of trials of random length. The schematic shows the three task-sets used in the recurrent session. The task-sets are non-overlapping from one episode to another in both the recurrent and the open-ended session, meaning that an episode switch produces a change of correct responses for all stimuli. (b) Proportion of correct responses to stimuli seen for the first time after the first correct response in an episode, during the last third of each experimental session. These newly seen stimuli are labeled *second* or *third* according to their order of appearance. Dots display the average for each subject. Violin plots display the shape of each distribution over subjects (Scott's rule). The black lines outline the mean ± s. e.m. (c) Performance preceding and following a trial with misleading feedback (non-rewarded correct response), at the end of episodes, averaged over all subjects (± s.e.m.). The subjects' performance did not change after a misleading feedback if it occured at the end of an episode, after being trained on the current task-set. (d) Illustration of the network model. The associative network (AN) is composed of a set of stimulus-selective populations and a set of action-selective populations. The synaptic weights between the two sets of populations are modified through a reward-modulated, activity-dependent Hebbian plasticity rule. At each trial, an action is selected via a soft and noisy winner-take-all mechanism with respect to the current set of synaptic weights. The task-set network (TN) is composed of neural populations selective to conjunctions of one stimulus and one action. Its activity is driven by the associative network's activity. The sequential activation of neural populations in the task-set network induces the potentiation of the synapses between them. An inference signal from the task-set network to the associative network biases the response to the stimulus on the next trial. (e) Illustration of the perfect, fully chunked encoding in the task-set network of the three non-overlapping task-sets from the recurrent session.

session (T-test on related samples, second stimulus: 0.53 ± 0.05 vs 0.39 ± 0.04, t = 2.1, p=0.049; third stimulus: 0.65 ± 0.05 vs 0.46 ± 0.03, t = 3.1, p=0.0049), demonstrating that subjects exploited information from the correct response to a given stimulus to infer the correct response to other stimuli. An important variability was however observed among subjects, as most of them did not learn task-sets perfectly, and some not at all (a point we return to later).

An additional observation consistent with task-set learning was that subjects do not modify their behavior following a misleading noisy feedback occurring late in an episode (*Figure 1c*, recurrent session: 0.94 ± 0.01 before misleading feedback, 0.94 ± 0.01 after misleading feedback, t = 0.25, p=0.80 ; open-ended session 0.94 ± 0.01 before, 0.93 ± 0.01 after, t = 0.68, p=0.50). An isolated misleading negative feedback after extensive learning in an episode should be ignored because inconsistent with the current task-set. A switch to another task-set or simply a change in a single stimulus-response association would be detrimental to performance. This negative feedback is indeed ignored by the subjects, indicating again they learn sets rather than individual stimulus-response associations.

## A network model for learning task-sets by chunking stimulus-response pairs

To examine the hypothesis that task-set-driven behavior emerges from unsupervised chunking of stimulus-response pairs, we studied an abstracted neural network model (*Figure 1d*), that built on previous modeling studies of a trace conditioning task in monkeys (*Fusi et al., 2007*; *Rigotti et al., 2010b*). While the model included some basic biological constraints, it was purposefully simplified to allow for straightforward fitting to human behavioral data on a subject-by-subject basis. The model consisted of two subnetworks, which we refer to as the Associative Network (AN) and the Task-set Network (TN). The associative network is a simplified version of a neural decision-making network (*Wang, 2002*; *Wong and Wang, 2006*; *Fusi et al., 2007*). It consists of a set of stimulus-selective populations and a set of action-selective populations, the activity in each population being for simplicity binary (active or inactive). The stimulus-action associations are learned through reward-modulated plasticity on the synapses between the two sets of populations (*Fusi et al., 2007*).

The task-set network consists of neural populations that display mixed-selectivity to all conjunctions of stimuli and actions (*Rigotti et al., 2010b*; *Rigotti et al., 2013*). For instance, if the associative network generates the action $A_2$ in response to the stimulus $S_1$, the corresponding population $S_1A_2$ is activated in the task-set network. Such mixed-selectivity response can be implemented through random projections from the associative network to the task-set network (*Rigotti et al., 2010a*; *Lindsay et al., 2017*), which for simplicity we don't explicitly include in the model. Synapses between neural populations in the task-set network undergo temporal Hebbian learning, that is they are modified based on the successions of stimulus-response pairs produced by the associative network (*Rigotti et al., 2010b*; *Ostojic and Fusi, 2013*). If two stimulus-response pairs are produced often after each other, the synapses between the corresponding mixed-selectivity populations in the task-set network are potentiated. When this potentiation exceeds a threshold (modeling in a simplified way recurrent inhibition), the two populations are chunked together. As a result, they are systematically co-activated when one of the two stimulus-response associations occurs. Thus, by means of temporal chunking, this subnetwork implements a task-set as a merged pattern of co-activated populations. This co-activation is communicated to the associative network, where it biases the stimulus-action associations at the next trial towards those encoded by the active populations in the task-set network. This effective *inference signal* helps the associative network determine the correct response to a stimulus different from the one in the current trial, and therefore implements task-set retrieval in the network model. To keep the model easy to fit and analyze, this inference signal is implemented in a simplified manner, by directly modifying the synaptic weights in the associative network (see Discussion for more realistic physiological implementations). The synaptic weights in the AN are therefore modified by a combination of sudden changes due to the inference signal and more gradual updates. The relative contribution of these two mechanisms is determined by a parameter that represents the strength of task-set retrieval (if it is zero, there is no retrieval). We will show that this specific parameter plays a key role in accounting for the variability in the subjects' behavior.

## Task-set encoding in the network model enables task-set driven behavior

The task-set network is in principle able to chunk together stimulus-response pairs that occur often after each other. We first show how it enables task-set-driven behavior. Consider an idealized situation at the end of the recurrent session of the experiment where full chunking has taken place, and the pattern of connectivity in the task-set network directly represents the three task-sets (*Figure 1e*).

Due to the inference signal from the task-set network to the associative network, this pattern of connectivity will directly influence the responses to incoming stimuli.

The impact of this inference signal is the strongest at an episode change, when the correct set of stimulus-response associations suddenly shifts (*Figure 2a,d,g*). The associative network always needs to discover the first correct association progressively by trial and error, by first depressing the set of stimulus-response synapses in the associative network corresponding to the previous task-set, and then progressively potentiating the synapses corresponding the new set of associations (*Figure 2a*). In the absence of task-set inference, this learning process happens gradually and independently for each stimulus (*Figure 2b*). In the presence of the idealized task-set network described above, once the first correct response is made, the task-set network produces the inference signal allowing the associative network to immediately modify its synaptic weight and recover the other two correct associations in the new episode (*Figure 2g,h*). As a consequence, the overall performance is increased (*Figure 2d*) due to a sudden increase in performance following the first correct response (*Figure 2e*).

A second situation in which the task-set inference strongly manifests itself is the case of noisy, misleading feedback late in an episode. At that point, the associative network haresoursecs fully learned the correct set of stimulus-response associations, and the performance has reached asymptotic levels. The network therefore produces mainly correct responses, but as on 10% of trials the feedback is misleading, it still occasionally receives negative reinforcement. In the absence of the task-set network, this negative feedback necessarily depresses the synapses that correspond to correct associations, leading to a decrease in performance on the following trials (*Figure 2c,f*). In contrast, in the presence of the idealized task-set network, the inference signal that biases the behavior towards the correct task-set is present despite the occasional negative feedback, and therefore allows the network to ignore it (*Figure 2c,f,i*). The encoding of task-sets in the task-set network pattern of connectivity therefore prevents the transient drop in performance, as seen in the experimental data (*Figure 1c*).

## Speed-accuracy trade-off for learning task-sets in the network model

The idealized encoding described above requires that the task-set network effectively and autonomously learns the correct pattern of connections corresponding to the actual task-sets. We therefore next examined under which conditions synaptic plasticity in the task-set network leads to correct learning, that is correct temporal chunking.

*Figure 3a,c,e,g* shows a simulation for a parameter set for which learning of task-sets proceeds successfully. At the beginning of the session, all populations within the task-set network are independent as all synaptic weights are below threshold for chunking. As the associative network starts producing correct responses by trial and error, the weights in the task-set network that connect correct stimulus-response pairs get progressively potentiated. While a fraction of them crosses threshold and leads to chunking during the first episode (and therefore starts producing the task-set inference signal), the majority does not, reflecting the fact that the first task-set is not fully learned at the end of the first episode. First, two stimulus-action associations are chunked together, then the third one is eventually added to the emerging cluster (*Figure 3—figure supplement 1*). The potentiation in the task-set network continues over several episodes, and the weights in the task-set network that correspond to co-occurring stimulus-response pairs eventually saturate to an equilibrium value. This equilibrium value is an increasing function of the probability that two stimulus-response pairs follow each other, and of the potentiation rate in the task-set network (see Materials and methods). The equilibrium synaptic weights in the task-set network therefore directly reflect the temporal contiguity between stimulus-response pairs (*Ostojic and Fusi, 2013*) and thus encode the task-sets. If the equilibrium value is larger than the inhibition threshold in the task-set network, this encoding will lead to the chunking of the activity of different populations and generate the inference signal from the task-set network to the associative network.

Learning in the task-set network is however strongly susceptible to noise and need not necessarily converge to the correct representation of task-sets. One important source of noise is the exploratory period following an episode switch, during which the associative network produces a large number of incorrect responses while searching for the correct one. If the potentiation rate in the task-set network is too high, the synaptic weights in the task-set network may track too fast the fluctuating and incorrect stimulus-response associations produced by the associative network, and quickly chunk

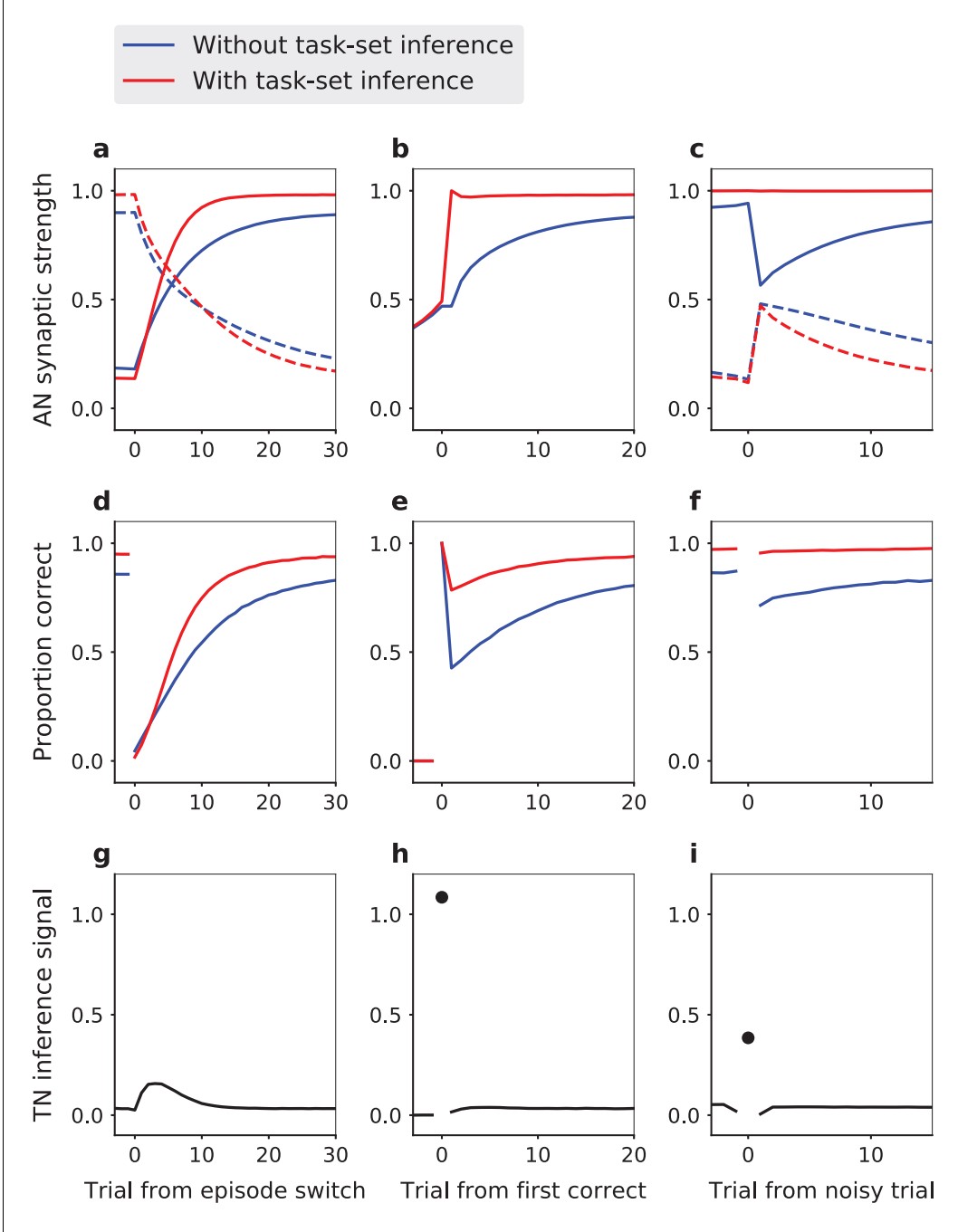

**Figure 2.** Task-set driven behavior in the network model with an idealized, perfect encoding of task-sets. The behavior of the model is compared in presence (red lines) and in absence (blue lines) of the inference signal from the task-set network, that allows task-set retrieval. (a,d,g) Model dynamics following an episode switch (at trial zero, the correct task-set shifts without explicit indication). (a) Strengths of synapses in the associative network between neural populations representing the new task-set (solid lines) and the previous task-set (dashed lines). (d) Performance (proportion of correct responses). (g) Mean change $<J_{INC} \cdot (1 - J^{AN})>$ in the AN synaptic weights due to the inference signal from the TN. Here the inference strength $J_{INC}$ is one, so that the weights in the AN reach their maximal values as soon as the network makes a first correct choice, and do not change afterwards. The first correct choice takes place randomly on different trials in different episodes leading to a spread over the first trials in the episodes and vanishing changes towards the end of the episode. (b,e,h) Task-set retrieval: same quantities as in (a,d,g), but aligned at the time of the first correct response. (c,f,i) Effect of misleading feedback: same

*Figure 2 continued on next page*

*Figure 2 continued*

quantities as in (**a,d,g**), aligned on a misleadingly non-rewarded correct trial at the end of episodes. Average of 5000 sessions of 25 episodes, with 10% of noisy trials. Network parameter values: $\alpha = 0.4$, $\beta = 7$, $\epsilon = 0$, $J_{INC} = 1$.

together pairs of events that do not correspond to a correct task-set (*Figure 3b*). Once these events are chunked together, the task-set network sends an incorrect inference signal to the associative network, and generates further incorrect associations (*Figure 3f,h*). As the network learns in an unsupervised fashion from its own activity, this in turn leads to more incorrect associations in the task-set network. In such a situation, the presence of the task-set network is at best useless and at worse detrimental to the performance of the network as a whole.

To determine under which conditions the plasticity in the task-set network leads to the correct learning of task-sets, we systematically varied the associative and task-set networks learning rates and compared the performance in the models with and without the inference signal from the task-set network. Our results show that the presence of task-set inference improves the network performance when the task-set network learning rate is slower than the associative network learning rate (red area in *Figure 4a*). As illustrated in *Figure 3b,d,f,h*, when learning in the task-set network is too fast, the network tracks noisy associations produced by the associative network, because of noise in the experimental feedback or because of errors made at the transition between episodes (blue area in *Figure 4a*). In contrast, slow learning allows the task-set network to integrate information over a longer timescale. While in principle it would be advantageous to learn the task-set structure as quickly as possible, the requirement to average-out fluctuations due to erroneous feedback sets an upper-bound on the learning rate in the task-set network. This is an instance of the classical speed-accuracy trade-off.

The correct learning of task-sets also depends on the strength of the inference signal. While strong inference leads to strong task-set retrieval and potentially large performance improvement, it also makes the network more sensitive to incorrect chunking in the task-set network. Our simulation show that larger inference strengths need to be compensated for by lower learning rates in the task-set network to produce an improvement in the performance (*Figure 4b*). This is another manifestation of the speed-accuracy trade-off.

## Fitting the model to behavioral data

Having described the dynamics in the model, we next proceeded to fit the model parameters to the subjects' behavioral data. In the full network model, we varied only five free parameters, which we determined independently for each subject by maximizing the likelihood of producing the same sequence of responses. To determine the importance of task-set retrieval, we compared the fit obtained from two versions of the model : the full model (associative network connected to the task-set network, five parameters), versus the associative network model alone, without the inference signal that allows for task-set retrieval (three parameters). In the open-ended session, in which a given task-set never reoccurs between episodes, the two models provided indistinguishable fits. In the recurrent session, the full model with task-set inference however provided a significantly better fit than the model without task-set inference (*Figure 5a*, Bayesian Information Criterion (BIC), T-test on related samples $t = 14$, $p = 4.3 \cdot 10^{-12}$; see also *Figure 5—figure supplement 1* for additional detail). In particular, the full model captured well the behavior at an episode change, where the subjects' performance exhibited a sudden increase following task-set retrieval at the first correct trial, combined with more gradual changes (*Figure 5c*).

Note that the models were fitted separately on the recurrent and open-ended sessions. Fitting the two sessions together led to a small but significant degradation of fitting performance (*Figure 5—figure supplement 3a*). More importantly, the models fitted simultaneously on both sessions were not able to capture the sudden increase in performance revealing task-set retrieval after the first correct trial (*Figure 5—figure supplement 3c*). This failure can be traced back to the need to adapt the learning rate in the task-set network between the two sessions, as previously observed for changes in task statistics (*Behrens et al., 2007*). Indeed, when the two sessions are fitted separately, the values of this learning rate strongly differ ($\overline{Q_P} = 0.17$ for the recurrent session versus $\overline{Q_P} = 0.44$ for the open-ended session, *Figure 5—source data 1*). In the recurrent session, on average over

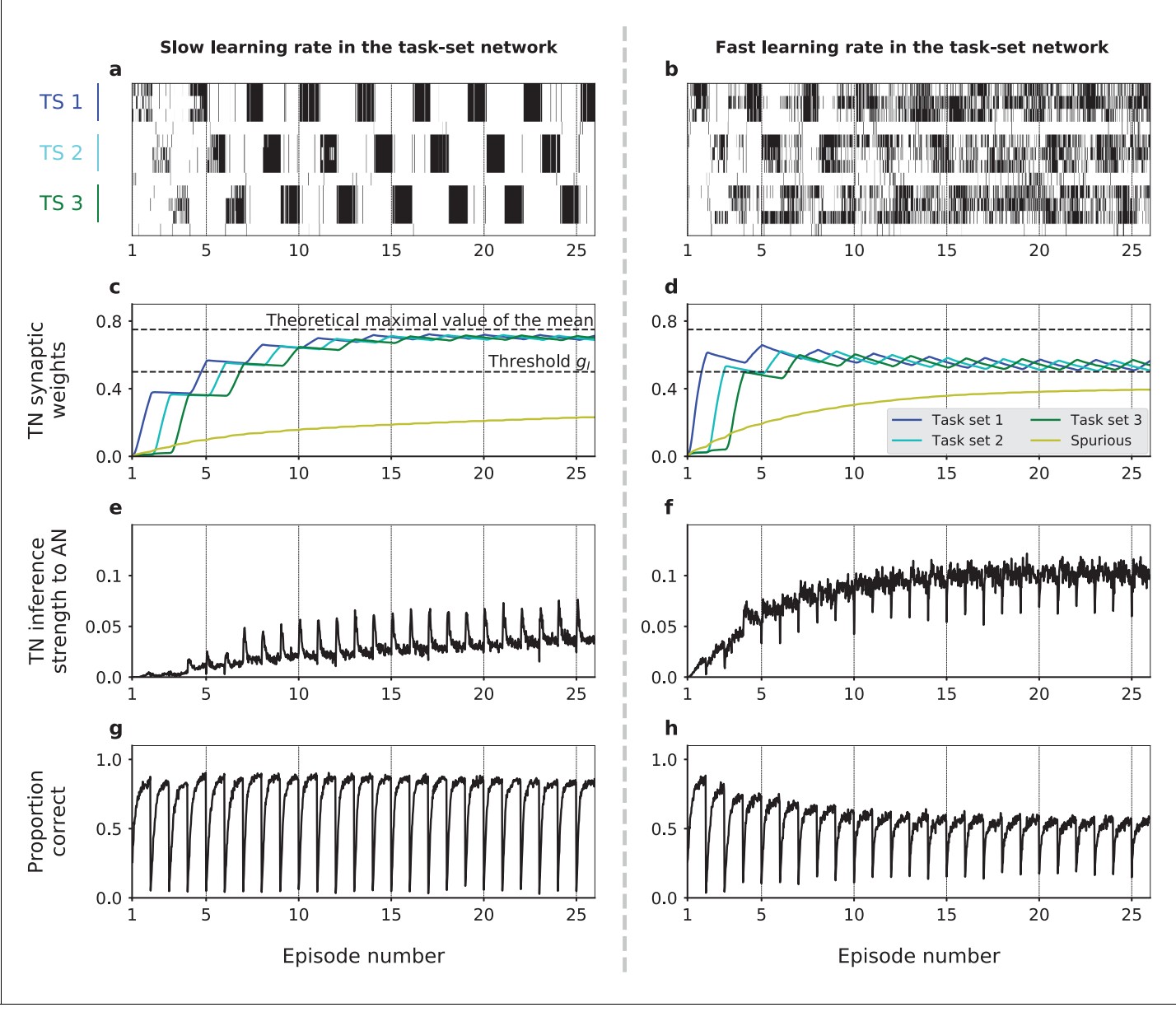

**Figure 3.** Dynamics of task-set learning. Left column: slow learning rate in the task-set network (TN) ($Q_P = 0.17$); right column: fast learning rate in the task-set network ($Q_P = 0.4$). (**a,b**) Activation of neural populations in the task-set network as a function of time during one session. In (**a**), learning dynamics proceed correctly and lead to the chunking of populations that correspond to the same task-set. As a result, the activation of one stimulus-response association causes the co-activation of the other two in the same task-set. In contrast, in (**b**) learning does not proceed correctly and chunking does not take place. (**c,d**) Average values of task-set network synaptic strengths between neural populations corresponding to each of the three correct task-sets, as well as 'spurious' synaptic strengths between neural populations from different task-sets or that do not correspond to any task-set at all. (**e, f**) Average value of the inference signal from the task-set network to the associative network connectivity. (**g,h**) Performance of the network. Task-sets presentation is periodic for illustration purposes. (**a,b**) corresponds to 1 run of the recurrent session. (**c,d,e,f,g,h**) corresponds to the average over 500 runs of the recurrent session. The values of parameters other than $Q_P$ were $\alpha = 0.4$, $\beta = 7$, $\epsilon = 0$, and $J_{INC} = 0.7$. .

The online version of this article includes the following figure supplement(s) for figure 3:

**Figure supplement 1.** The chunking of 3 stimulus-action associations into a single task-set is gradual.
**Figure supplement 2.** Learning overlapping task-sets.

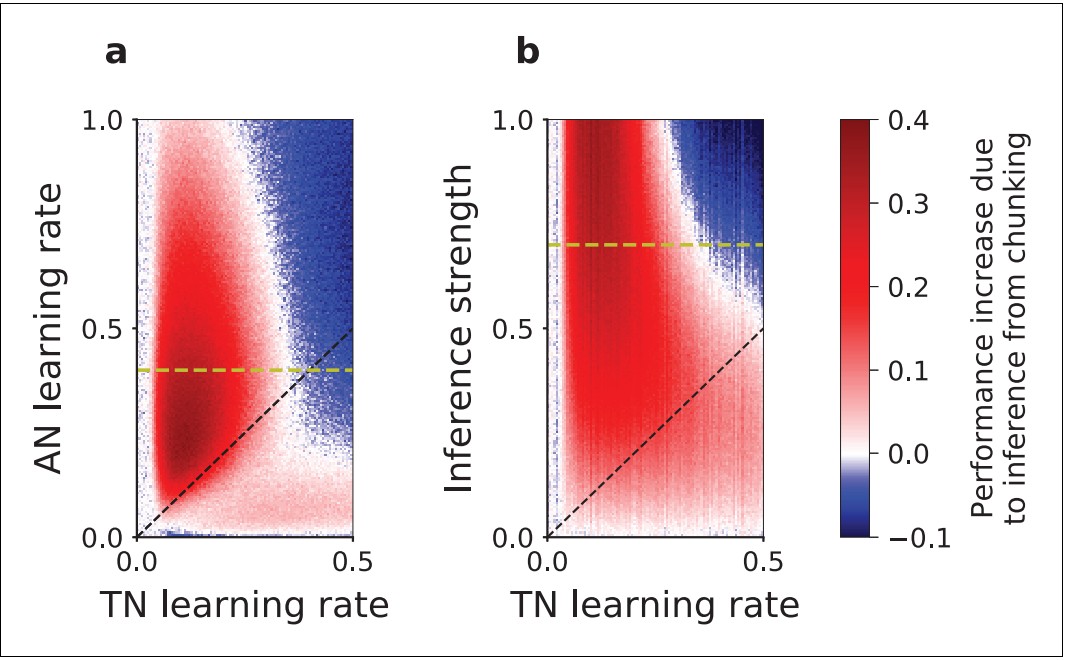

**Figure 4.** Slow versus fast learning: conditions for correct encoding of task-sets in the network model. **(a)** Difference in the performance of the network model with or without task-set inference, plotted as a function of the associative network learning rate $\alpha$ and the task-set network learning rate $Q_P$, (with $\beta = 7$ and inference strength $J_{INC} = 0.7$). **(b)** Same difference in performance but plotted as a function of the inference strength $J_{INC}$ and the task-set network learning rate $Q_P$, (with $\beta = 7$ and associative network learning rate $\alpha = 0.4$). We computed the performance averaged over the five first correct responses for a stimulus, in the last third of the session, on an average of 200 runs of the recurrent session and with 10% noisy trials. The dashed black lines mark the diagonal. The dashed yellow lines correspond to $\alpha = 0.4$ and $J_{INC} = 0.7$ respectively, and relate **(a)** to **(b)**.

subjects, the learning rate in the task-set network ($\overline{Q_P} = 0.17$, $\sigma = 0.0070$) is half of the learning rate in the associative network ($\overline{\alpha} = 0.35$, $\sigma = 0.0073$), which is consistent with our initial prediction that the learning rate in the task-set network needs to be slower than in the associative network.

As mentioned earlier, an important behavioral variability was present among subjects. This variability was particularly apparent in the performance following an episode switch, where some subjects increased their performance much faster than others in the recurrent session, compared to the open-ended session (*Figure 1b*). Inspecting the parameter values obtained for different subjects revealed that the most variable model parameter between subjects (*Figure 5—source data 1*) was the strength of the inference signal for task-set retrieval in the model (*Figure 5b*, T-test on related samples $t = 14.8$, $p = 1.5 \cdot 10^{-12}$). This parameter directly quantifies the strength of the sudden change in performance corresponding to task-set retrieval at the start of an episode. The value of this parameter appeared to directly account for the inter-subject variability, as it correlated with the difference between BIC values obtained for models with and without task-set inference (linear regression $r^2 = 0.81$, $p = 1.4 \cdot 10^{-8}$, *Figure 5d*) as well as with the subjects' performance following the first correct trial in an episode (linear regression $r^2 = 0.60$, $p = 2.5 \cdot 10^{-5}$, *Figure 5e*). These findings further suggest the variability in that parameter is directly linked with the subject's ability to recover task-sets. This was confirmed by examining the results of a behaviorally-independent post-test debriefing used in the original study to classify subjects as either exploiting task-set structure ('exploiting' subjects) or not ('exploring' subjects) (see Materials and methods and *Figure 5d,e*). Exploiting subjects systematically corresponded to higher performance on trials following a correct response (T-test $t = 4.9$, $p = 9.7 \cdot 10^{-5}$) and higher values of the inference parameter in the model (T-test $t = 2.9$, $p = 9.1 \cdot 10^{-3}$, see also *Figure 6—figure supplement 3*).

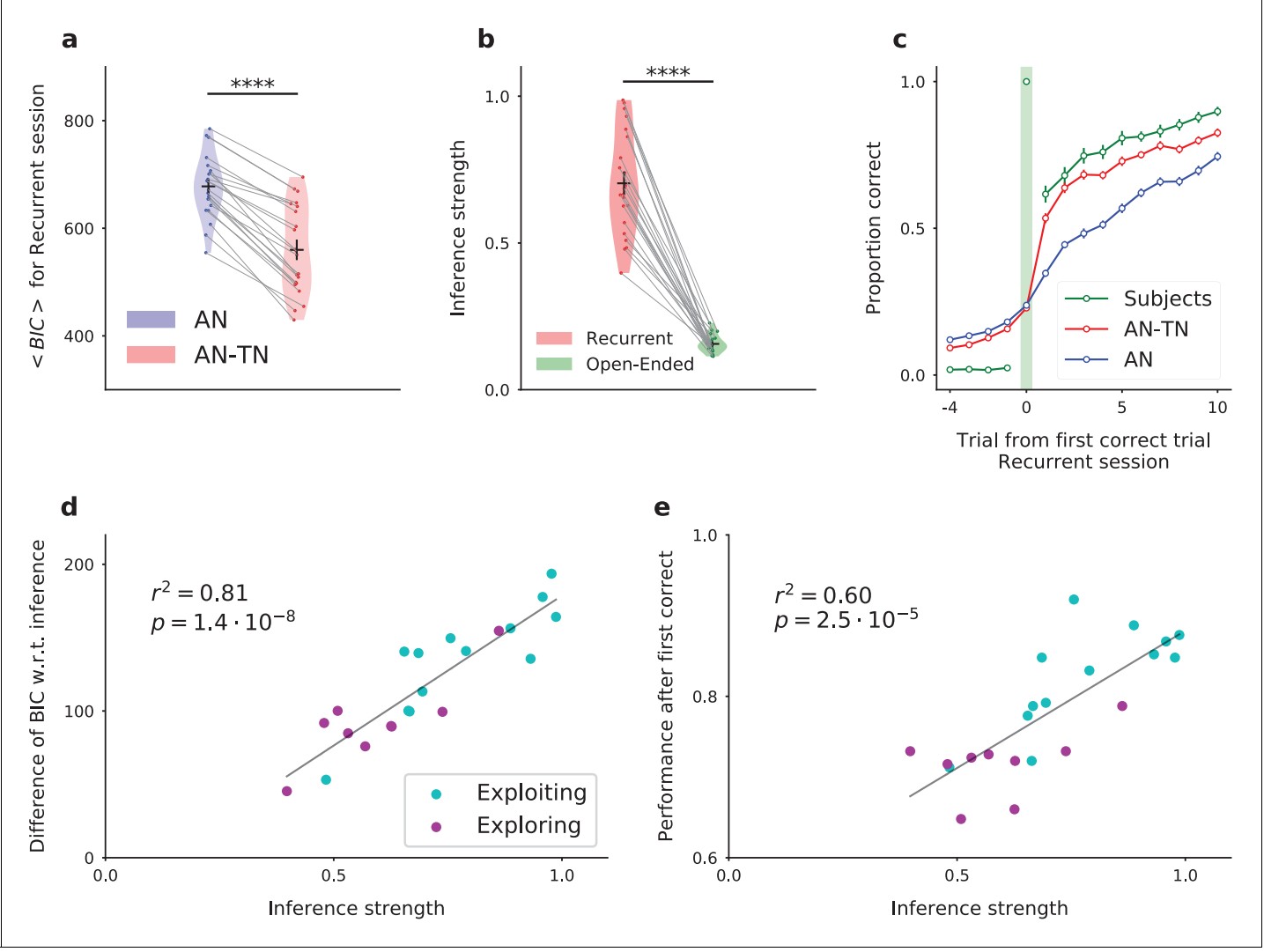

**Figure 5.** Fitting the model to experimental data: the model with inference (AN-TN) captures the statistical structure of the data, and accounts for the variability between subjects. (**a**) Model comparison for the recurrent session. Bayesian Information Criterion (see Materials and methods) for the models with and without task-set inference. The model provides a significantly better fit with inference than without. (**b**) Estimate of the inference strength $J_{INC}$ from the task-set network to the associative network connectivity in the model with task-set inference, for both sessions. (**c**) Proportion correct around the first correct trial, averaged over episodes and over subjects, for the recurrent session. (**d**) Subject by subject difference between BIC values obtained for models with and without task-set inference, as a function of the inference strength parameter, for the recurrent session. Subjects are classified as 'exploiting' or 'exploring' from a post-test debriefing. The grey line displays a least-squares regression. (**e**) Subject by subject performance following the first correct trial in an episode, as a function of the inference strength parameter, for the recurrent session. The performance was computed by considering the 10 trials following the first correct trial of each episode. The grey line displays a least-squares regression. .

The online version of this article includes the following source data and figure supplement(s) for figure 5:

**Source data 1.** The table summarizes the full network (AN-TN, with inference) and the associative network alone (AN, without inference) models fitting performances and average parameters.

**Figure supplement 1.** Model comparison for the recurrent session.

**Figure supplement 2.** Learning task-sets with a lower ratio of potentiation versus depression in the task-set network ($Q_P/Q_M = 5$) by refitting the model, while either fixing $g_I = 0.5$ or $g_I = 0.2$.

**Figure supplement 3.** Model fit for both sessions together.

## Testing model predictions for task-set retrieval

We next examined a specific subset of experimental trials where task-set retrieval is expected to take place. In the model, how quickly two stimulus-response pairs are chunked together depends on how often they co-occur, as well as on the value of the learning rate in the task-set network. Once

two pairs are chunked together, the correct response to the stimulus corresponding to one of the pairs leads to the retrieval of the task-set and biases the response to the stimulus from the second pair. When the pairs are not chunked together, the responses to the two stimuli are instead independent. The basic prediction is therefore that the responses to the stimulus from the second pair should differ between trials when chunking has or has not taken place, depending on the learning progress.

We first tested this prediction in a situation where chunking should lead to the retrieval of the correct task-set. We focused on one trial in each episode, the trial that followed the first correct response (*Figure 6a*), for a different stimulus. Running our model on the full sequence of preceding experimental events (on a subject-by-subject basis, using parameters fitted to each subject and actual sequences of stimuli and responses) produced a prediction for whether chunking had occurred for this trial (*chunked* or *independent*, *Figure 6a*, orange and grey respectively). The model with inference predicted that the proportion of correct responses on chunked trials should be higher than on independent trials due to the inference signal implementing task-set retrieval. In the model without inference where the associative network is independent of the task-set network, the performance on the two types of trials is instead indistinguishable. Comparing the proportion of correct responses on experimental trials classified in the two categories showed a significant increase for chunked trials compared to independent trials (*Figure 6b*: (i) model without inference $t = 0.64$, $p = 0.53$, (ii) model with inference $t = 6.9$, $p = 1.4 \cdot 10^{-7}$, (iii) subjects $t = 2.8$, $p = 8.8 \cdot 10^{-3}$, (iv) chunked trials, model without inference versus model with inference $t = 11$, $p = 6.3 \cdot 10^{-10}$, (v) chunked trials, model without inference versus subjects $t = 5.9$, $p = 2.3 \cdot 10^{-5}$), so that the model prediction with inference was directly borne out by experimental data. The task-set retrieval predicted by the model therefore led to a clear increase of subjects' performance. Moreover, reaction times on chunked trials were significantly lower than on independent trials, showing that the inference helped subjects to be faster at responding (*Figure 6c*, $t = 8.7$, $p = 1.7 \cdot 10^{-9}$). This provides a supplementary validation, as the model was not fitted on reaction times. The model additionally predicted a sudden switch in behavior after two stimulus-response associations were chunked together. Splitting the data of *Figure 6* as a function of episode number revealed a pattern consistent with such a switch (*Figure 6—figure supplement 2*). Note that a potential confound could be induced if the chunked trials appeared on average later in an episode than independent trials. A direct comparison however showed that the distributions of chunked and independent trials were indistinguishable (*Figure 6—figure supplement 1a*, $ks = 0.085$, $p = 0.62$).

We next tested the predictions of the model on trials where chunking leads to the retrieval of an incorrect task-set. Such a situation happens because of the presence of 10% of trials with misleading feedback, which may indicate to the subject that the produced response was correct although it was not. Our model predicted that in this case incorrect task-set retrieval leads to a decrease of the performance on the next trial. To test this prediction, we first detected the misleading trials, and then used the model to classify each of them as either chunked or independent (*Figure 6d*). Comparing in the experimental data the responses on chunked trials with the performance of the model without task-set inference showed that indeed the subjects' performance was significantly reduced when the model predicted an incorrect task-set retrieval (*Figure 6e*: (i) chunked trials, model without inference versus model with inference $t = 5.8$, $p = 1.0 \cdot 10^{-5}$, (ii) chunked trials, model without inference versus subjects $t = 5.2$, $p = 2.2 \cdot 10^{-5}$).

The two behaviors described above (retrieval of a correct task-set after the first correct response, and retrieval of an incorrect task-set after a misleading feedback) cannot be predicted by the model without inference: we thus assessed the generative performance of our chunking mechanism and *falsified* the model without inference (*Palminteri et al., 2017*).

## Neural correlates of task-set inference

Using the model fitted to individual subjects, we next aimed to identify the neural correlates of task-set inference based on blood-oxygen-level-dependent (BOLD) signal recorded from functional magnetic resonance imaging (40 subjects, Experiment 2 [*Donoso et al., 2014*], see *Figure 5—figure supplement 1a* for model fits on this dataset). We specifically examined the correlations between the task-set inference and the BOLD signal at the time of feedback, while controlling for trial difficulty and prediction error (see Materials and methods). In the recurrent session (*Figure 7—source*

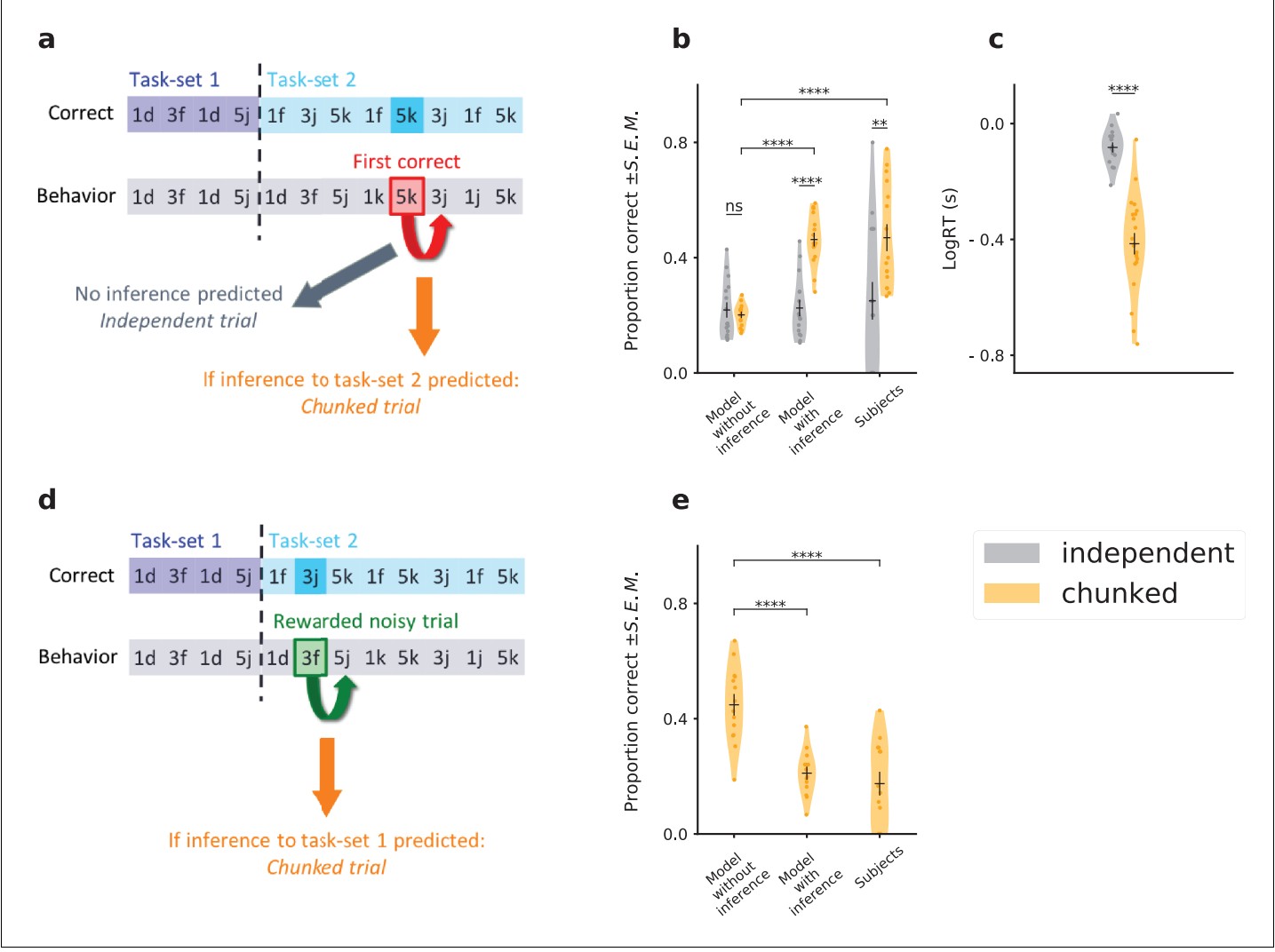

**Figure 6.** Testing the predictions of the temporal chunking mechanism on specific trials. (a) Schematic of the prediction for correct task-set retrieval. For each episode switch, and subject by subject, we compute the probability of making a correct choice after the first correct trial, for a different stimulus. Trials are classified from a model-based criterium as 'chunked' or 'independent', respectively depending on the presence or absence of an inference from the task-set network to the associative network. (b) Because of task-set inference, the model predicts a significant increase of performance on chunked trials compared to independent trials. This is not predicted by the associative network alone ('Model without inference'). Subjects' performance on these trials matches the model with inference. The error bars are larger for the *independent* trials because this category contains half the amount of data, as shown in *Figure 6—figure supplement 1*. (c) Log of subjects' reaction times in seconds, for trials classified as chunked or independent. (d) Schematic of the prediction for task-set retrieval following misleading rewarded trials. After each episode switch, the subject makes incorrect choices. On 10% of these trials the feedback is misleadingly rewarded (e.g. 3*f*, which corresponds to a correct association for the previous task-set, but not for the current task-set). Because of the inference from the task-set network, the previous task-set can be incorrectly inferred by the model from the misleading reward. (e) Probability of a correct association after a misleadingly rewarded noisy trial classified as a chunked trial by the model. The model with inference predicts an incorrect association at the next trial, producing a decrease in performance. This decrease is not predicted by the associative network alone ('Model without inference'). Subject's performance on these trials matches the model with inference. Violin plots display the shape of each distribution (Scott's rule). Dots display the average for each subject. The black lines outline the mean ± s.e.m. .

The online version of this article includes the following figure supplement(s) for figure 6:

**Figure supplement 1.** Task-set retrieval prediction.

**Figure supplement 2.** Testing the predictions of the temporal chunking mechanism as learning evolves.

**Figure supplement 3.** Histograms over subjects of the difference of performance after five first consecutive correct trials, between the recurrent session and the open-ended session.

*data 1*, bottom), BOLD activity correlated positively with the inference signal strength in dorsolateral prefrontal cortex, dorsal anterior cingulate cortex and anterior supplementary motor area; and negatively in ventromedial prefrontal cortex. In contrast, in the open-ended session, we found no significant positive or negative effect in frontal lobes corresponding to this parametric modulator.

To further investigate the difference between the recurrent and open-ended sessions, we focused on a specific set of regions of interest (ROIs), defined based on significant BOLD activations in both sessions for the task-set inference signal (see *Table 1* and Materials and methods). These ROIs consisted of voxels in the ventromedial, dorsomedial and dorsolateral prefrontal cortex (respectively left, middle, and right columns of *Figure 7a*). As they were defined based on the activity in both sessions considered together, these ROIs did not promote differences between the two sessions. However, we found that the correlation of the task-set inference signal with BOLD activity in dorsomedial and dorsolateral prefrontal cortex was significantly stronger in the recurrent session than in the open-ended session (dmPFC: $t = 3.0$, $p = 3.2 \cdot 10^{-3}$ ; and dlPFC: $t = 4.6$, $p = 1.6 \cdot 10^{-5}$), while activations in ventromedial prefrontal cortex did not discriminate significantly between the two sessions ($t = 0.16$, $p = 0.87$). This analysis showed a difference of neural activity in the dorsal system corresponding to the necessity of learning and using (recurrent session) or not (open-ended session) the model of the task. Additional controls were performed using ROIs selected from several meta-analyses (*Lancaster et al., 1997*; *Yarkoni et al., 2011*; *Shirer et al., 2012*; *Glasser et al., 2016*, see Materials and methods and *Figure 7—source data 2*). Altogether, they confirmed dorsolateral and dorsomedial prefrontal cortex were specifically recruited in the recurrent session and correlated with task-set inference.

Since our model predicted the specific trial at the beginning of an episode where task-set inference should be maximal, we finally compared the BOLD activity on that trial with two trials immediately before and after it. Using the same ROIs as defined above, we found that they did not exhibit a response in the trials immediately preceding the predicted task-set retrieval. In contrast, dorsomedial responses were significant at task-set retrieval, and ventromedial and dorsolateral responses were significant from this trial onwards (*Figure 7b*). The specific trial predicted by the model for task-set retrieval in each episode was thus confirmed both behaviorally (*Figure 6*) and neurally (*Figure 7*).

## Discussion

In this study, we tested a model in which task sets emerged through unsupervised temporal chunking of stimulus-action associations that occurred in temporal contiguity. When repeated, a task-set could be retrieved from a single stimulus-action association by reactivation of the whole chunk. This retrieval then biased the following stimulus-response association through an inference signal to the decision-making circuit. The model predicted abrupt changes in behavioral responses on specific trials in a task-set learning experiment (*Collins and Koechlin, 2012*; *Donoso et al., 2014*). Testing these predictions, we showed that the retrieval of a task-set had both adaptive (reduction of exploration) and sometimes maladaptive effects (retrieval of an incorrect task-set) on the following trial performance. Our analysis of BOLD activity established a functional network engaging ventromedial, dorsomedial, and dorsolateral prefrontal cortex that correlated with the inference signal for task-set

**Table 1.** One-way ANOVA defining the regions of interest used for the analysis of BOLD correlates of the task-set inference signal. The ROIs are defined from activations from the parametric modulator corresponding to the TN inference signal, in both sessions (contrasts REC+OE and -REC-OE, FWE $p = 0.05$). dlPFC: dorsolateral prefrontal cortex; dmPFC: dorsomedial prefrontal cortex; vmPFC: ventromedial prefrontal cortex; [x y z] are MNI coordinates; REC: Recurrent session; OE: Open-Ended session.

| Contrast | Label | [x y z] | Brodmann areas | Glasser parcellation | t-value | Cluster size |
|---|---|---|---|---|---|---|
| REC+OE | right dlPFC | [32 12 60] | 6,8,9,10,11,44,45,46 | 6sma, 8av, 8C, p9-46v, 46, a9-56v, 9-46d, 9a, i6-8, s6-8 | 9.25 | 519 |
| | left dlPFC | [−48 4 36] | 6,8,9 | 8Av, 8 c | 6.27 | 25 |
| | dmPFC | [4 24 48] | 6,8,9,32 | SFL, SCEF, p32pr, d32, 8BM, 8BL, a32pr | 7.86 | 73 |
| Negative(REC+OE) | vmPFC | [−12 48–4] | 9,10,11,32 | a24, d32, p32, 10 r, 9 m, 9 p, 9a, 10 v, 25, s32, p24 | 7.64 | 225 |

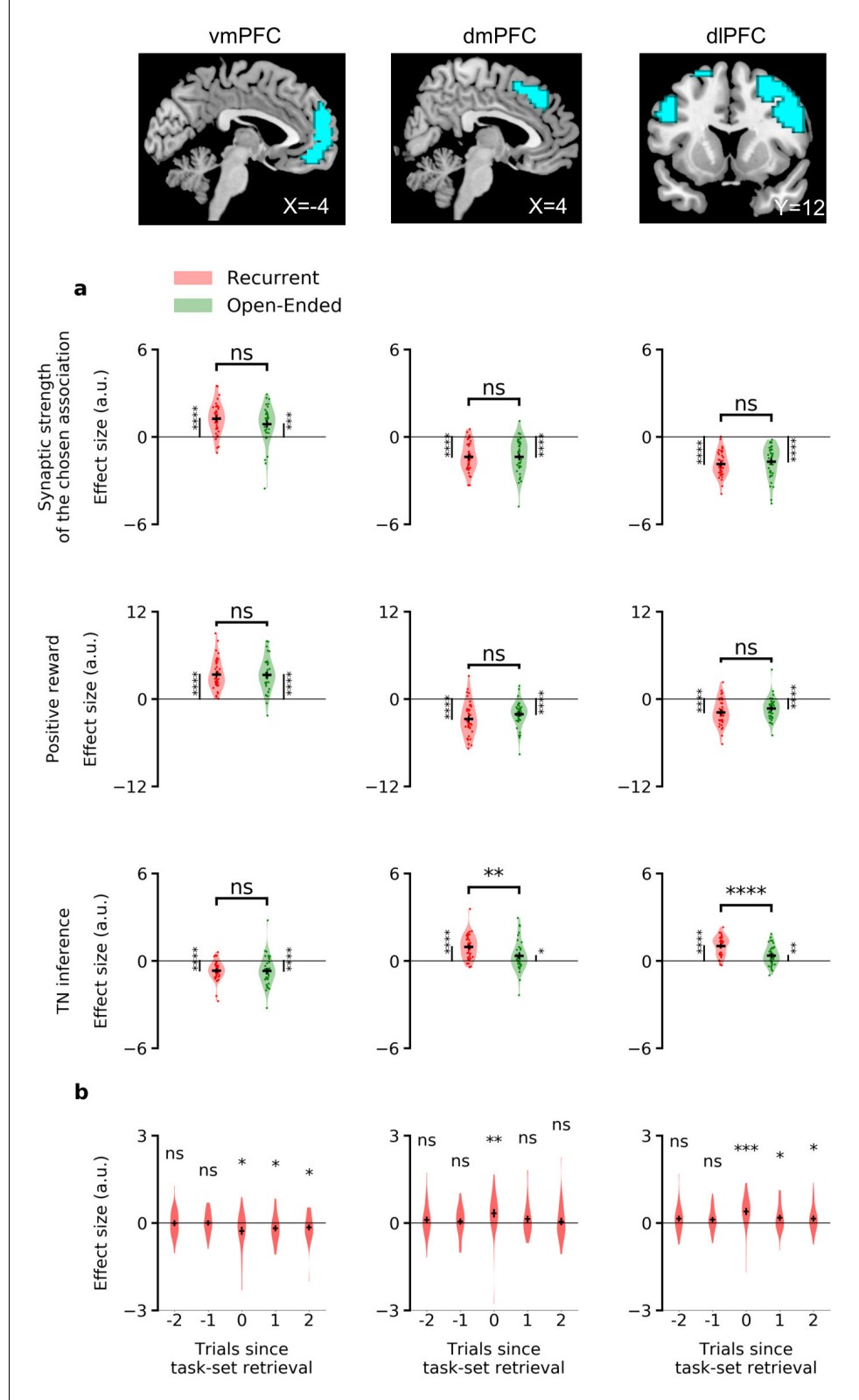

**Figure 7.** ROI analyses of the neural correlates of task-set inference. The areas in blue represent the regions of interest identified in the previous analysis (*Table 1*) using a significant threshold of FWE $p = 0.05$. (a) Correlations between the BOLD signal at the onset feedback and the parametric modulators of the time series of $W_{chosen}$, positive rewards, and the inference signal. (b) Comparison of BOLD activity on the first chunked trial of the model

*Figure 7 continued on next page*

*Figure 7 continued*

behavioral predictions (per episode, if it existed from sufficient learning, *Figure 6a,b,c*) with two trials immediately before and after it (see Materials and methods). Effect sizes in arbitrary units for the recurrent and the open-ended session. Error bars correspond to the standard error of the mean over the 40 subjects. dlPFC: dorsolateral prefrontal cortex; dmPFC: dorsomedial prefrontal cortex; vmPFC: ventromedial prefrontal cortex .

The online version of this article includes the following source data and figure supplement(s) for figure 7:

**Source data 1.** Neural correlates of the synaptic strength in the associative network, and of the inference from the task-set network to the associative network.

**Source data 2.** Control independent ROI analysis: neural correlates of the inference signal from the task-set network to the associative network, at the onset feedback.

**Figure supplement 1.** ROI analysis in the hippocampus.

**Figure supplement 2.** TN inference parametric regressor for four selected subjects, for the recurrent session (left) and the open-ended session (right).

---

retrieval. The dorsal system was engaged preferentially in the situation where the retrieval of a task-set was used to improve performance.

Previous computational models of task-set-based behavior fall into two broad categories (*Frank and Badre, 2012*). On the one hand, behavioral and task-set learning are modeled on an abstract, psychological level (*Botvinick et al., 2009*; *Collins and Koechlin, 2012*; *Donoso et al., 2014* ). Models of this type can be directly used to fit subjects' behavior and correlate abstract variables with BOLD activity, however they do not address the underlying biological mechanisms. On the other hand, task-set-based behavior and learning have been modeled using detailed, biologically-inspired networks (*O'Reilly, 1998*; *O'Reilly and Munakata, 2000*; *Durstewitz et al., 2000*; *O'Reilly, 2006*; *O'Reilly and Frank, 2006*; *Frank and Badre, 2012*; *Collins and Frank, 2013*). These models typically include a large number of free parameters, and are difficult to fit to the available data, so that directly testing the underlying biological mechanisms is challenging. Here we adopted an approach that interpolates between these two extremes, and used an intermediate-level model, which was based on biologically constrained models, but highly abstracted to allow for direct comparison with human behavioral data. This model was moreover developed to test a specific hypothesis, namely that task-set learning relies on Hebbian chunking of individual stimulus-response associations.

## Biologically plausibility of the temporal chunking mechanism

While the network dynamics and plasticity mechanisms in the network model were highly abstracted, they nevertheless included some basic biological constraints present in more detailed models. The reward-dependent Hebbian learning and winner-take-all mechanisms in the associative network were based on previous studies in the field of conditional associative learning (*Williams, 1992*; *Wang, 2002*; *Wong and Wang, 2006*; *Fusi et al., 2007*; *Soltani and Wang, 2010*; *Soltani et al., 2017*; *Hertz, 2018*; *Sutton and Barto, 2018*). The temporal chunking mechanism in the task-set network was based on Hebbian plasticity at behavioral timescales, and can be generated for instance through sustained neural activity and extended STDP (*van Rossum et al., 2000*; *Compte et al., 2000*; *Miller and Cohen, 2001*; *Rougier et al., 2005*; *Drew and Abbott, 2006*; *Curtis and Lee, 2010*; *Murray et al., 2017*). The required mixed-selectivity can be obtained from randomly connected neurons receiving feed-forward inputs coming from sensory and motor areas and has been widely observed in the prefrontal cortex (*Asaad et al., 1998*; *Wallis et al., 2001*; *Genovesio et al., 2005*; *Rigotti et al., 2013*).

Despite the basic biological plausibility of the underlying mechanisms, implementing task-set learning in a more detailed model with heterogeneous connectivity and continuous-valued, or spiking neurons driving ongoing plasticity would remain a challenge (see e.g. [*Zenke et al., 2015*]). A particularly difficult point lies in implementing the inference signal for task-set retrieval, which essentially solves an exclusive-or gating problem, as different task-sets map from the same set of stimuli onto different actions (*Rigotti et al., 2010a*). In our model, we have used a highly simplified implementation where the inference directly gates the synaptic weights in the associative network. An alternative approach could be to include additional intermediate layers of neurons, and gate the activity of neural populations in these intermediate layers. Previous works have relied either on

highly structured layers representing different brain regions (*Frank and Badre, 2012*; *Collins and Frank, 2013*) or on randomly connected layers of non-linear mixed selective cells (*Rigotti et al., 2013*; *Fusi et al., 2016*). Other possibilities include implementing contextual gating in a single, recurrent network through mixed, low-rank connectivity (*Mastrogiuseppe and Ostojic, 2018*), or specialized neural sub-populations (*Yang et al., 2019*; *Dubreuil et al., 2019*). Ultimately, a biologically plausible instantiation of this gating remains an open question, as the currently available physiological data does not provide sufficient constraints.

## Neural correlates of task-set retrieval

Consistent with the literature on the neural correlates of goal-directed behavior, we found that the inference signal for task-set retrieval correlated specifically with BOLD activity in ventromedial, dorsomedial and dorsolateral prefrontal networks. In particular, the ventromedial prefrontal cortex correlated negatively with the inference signal, that is positively with the compatibility between encoding in the two subnetworks when a reward was received (as a prediction error-like signal). This is potentially in accordance with the role of ventromedial prefrontal cortex in monitoring the Bayesian actor reliability in this experiment (*Donoso et al., 2014*). The dorsal prefrontal cortex was preferentially engaged when the model of the task (i.e., the task-sets) is useful and integrates into the behavioral policy (recurrent versus open-ended sessions, while controlling for trial perceived difficulty, as implemented by reaction times [*Shenhav et al., 2013*; *Shenhav et al., 2014*]). Dorsolateral prefrontal cortex is known to be specifically engaged for temporally integrating and organizing multimodal information (*Miller, 2000*; *Duncan, 2001*; *Sakai, 2008*; *Kim et al., 2008*; *O'Reilly, 2010*; *Ma et al., 2014*). Previous work showed that neurons in the anterior cingulate cortex monitor the allocation and the intensity of control (*Dosenbach et al., 2006*; *Behrens et al., 2007*; *Rushworth et al., 2007*; *Shenhav et al., 2013*; *Khamassi et al., 2013*; *Enel et al., 2016*). In this specific experiment, using a Bayesian framework, the dorsal anterior cingulate cortex was shown to be specifically selective to switch-in events (*Donoso et al., 2014*).

## Predictions for experiments with overlapping task sets

In the experiments we modeled, the different task-sets in the recurrent session consisted of fully distinct sets of stimulus-action associations. In a more complex situation, two different task-sets could partly overlap by sharing a common association. Our model makes interesting predictions for this setup. We studied two cases: the case where a newly introduced task-set partly overlaps with a previously learned one, and the case where two task-sets partially overlap from the beginning of learning. In both cases, our model predicts that the overlapping association induces a decrease in performance on the following trial. In the first case, this decrease is transient, while in the second case it is permanent..

For the first case (*Figure 3—figure supplement 2a,b,c,d,e*), we simulated the recurrent session till episode 25, as in *Figure 3a,c,e,g*. After episode 25, we introduced a fourth task-set (task-set 4, see *Figure 3—figure supplement 2a*) that had one overlapping stimulus-action association with task-set 1 (association [5 j]). When this new task-set is introduced, in the task-set network the overlapping association [5 j] is chunked with stimulus-response associations corresponding to task-set 1. In consequence, any trial on which stimulus five is shown leads to an incorrect retrieval of task-set 1, and therefore errors on the next trial (*Figure 3—figure supplement 2b*). This is analogous to maladaptive retrieval examined in *Figure 6d,e*. Synaptic depression in the task-set network eventually breaks away the association [5 j] from the cluster corresponding to task-set 1, and chunks it with task-set 4, at which the performance on the trials following stimulus five increases.

For the second case, we considered (in *Figure 3—figure supplement 2f,g*) a session with three task-sets, among which two shared the same stimulus-response association (task-set one and task-set 3, for the association [5 j], see *Figure 3—figure supplement 2f*). In the model, the shared association becomes either chunked with both task-sets, or with neither of the two, but is in either case uninformative on the correct task-set. The model therefore predicts a decrease of performance on the trial following the shared association, in particular at the beginning of an episode (*Figure 3—figure supplement 2g*).

## Stability/flexibility trade-off from the unsupervised temporal chunking mechanism

Our model builds on an attractor chunking mechanism (*Rigotti et al., 2010b*) while being simplified: we don't make the hypothesis of the existence of fixed attractors. Instead, the synaptic weights are modified immediately from the start and continuously. Thus, the task-set network can learn from its own activity, combining prior statistical information to future learning, crucial in non-stationary problems. In order for this mechanism to be stable when learning concurrent task-sets, learning has to be slower as the representational complexity increases.

This mechanism also enables the encoding of a synaptic trace of any sequence of events, even incorrect, as a transition probability (weak but non-zero) between chunks or with an isolated neural population. The brain relies on estimates of uncertainty within and between task-sets (*Yu and Dayan, 2005*; *Courville et al., 2006*; *Collins and Koechlin, 2012*; *Rushworth and Behrens, 2008*; *Gershman et al., 2012*; *Kepecs and Mainen, 2012*; *Donoso et al., 2014*; *Soltani and Izquierdo, 2019*). More specifically, *Collins and Koechlin (2012)* have shown that the Bayesian likelihood ('reliability') of each task-set in memory is evaluated. Inferences on the current and alternative task-sets have been found to occur in medial and lateral prefrontal cortices respectively *Donoso et al. (2014)*. The coupling between these two tracks permits hypothesis testing for optimal behavior. Future work could investigate how an estimate of uncertainty (or reliability over task-sets) is retrieved from the synaptic weights of our model.

## Temporal chunking as a mechanism for hierarchical learning, multi-step transition maps and generalization

The model used in this study relies on two network layers that operate on separate timescales, with the task-set network learning the statistics of the associative network on a slower timescale. More generally, cognitive control and learning depend on a succession of hierarchical representations in the brain (*Badre, 2008*; *Badre et al., 2009*; *Badre et al., 2010*). Plasticity and chunking between mixed-selective cells may therefore create a conjunctive code seeding a flexible 'representational medium' (*Miller, 2000*; *Duncan, 2001*; *Ma et al., 2014*; *Manohar et al., 2019*). Temporal Hebbian learning on a hierarchy of different timescales may create chunks combining states, action, rewards, or more abstract 'task-sets' to create hierarchically more complex representations. Moreover, this type of plasticity provides a potential mechanism for learning transition probabilities on this complex state-space, and therefore suggests a candidate implementation the successor representation (with the augmentation of eligibility traces, [*Gershman and Niv, 2012*; *Russek et al., 2017*]) and multi-step transitions maps (*Kahana, 1996*).

This mechanism can also lead to generalization. Augmenting our network with a generalization layer composed of neurons selective to the combination of three stimuli and three actions could produce faster learning of a new task-set by biasing lower cortical structures. In this simplistic scheme, generalization is a top-down, gating-like mechanism solving exclusive-or problems between layers of cells of decreasing complexity. Caching multi-steps transitions in a single value (model-free) or not (model-based) would then be equivalent to learning at slower timescales in an increasingly complex hierarchy of cortex layers (*Murray et al., 2014*).

## Materials and methods

### Experimental procedures

#### The experimental task

We modeled a specific human experiment for concurrent task-set monitoring, previously reported in *Collins and Koechlin (2012)*; *Donoso et al. (2014)*. The detail of the experimental procedures can be found in the original papers, here we provide only a summary. Data from *Collins and Koechlin (2012)* are called *Experiment 1*. Data from *Donoso et al. (2014)* are called *Experiment 2*. The experimental designs are identical. *Experiment one* is a behavioral experiment including 22 subjects. *Experiment 2* involves 40 subjects, with fMRI acquisition.

Subjects had to search for implicit associations between 3 digits and four letters by trial and error. In each trial, a visual stimulus (a digit on the screen in {1, 3, 5} or {2, 4, 6}) was presented to the subject (*Figure 1a*). The subject had to take an action by pressing a letter on a keyboard in {d, f, j, k}.

The outcome (reward or no reward) was announced with a visual and auditory feedback. A visual measure of the cumulative collected profit was displayed on the screen. For each trial of Experiment 1, the subject had 1.5 s to reply during the presentation of the stimulus. The average length of a trial was 2.9 s. For Experiment 2, the mean of a trial was either 6 s or 3.6 s, depending on whether BOLD activity is acquired or not. MRI trials were longer, introducing jitters at stimulus or reward onsets for signal decorrelation.

A *correct association* between the stimulus and the action led to positive reward with a probability 90%. An incorrect association between the stimulus and the action led to a null reward with a probability 90%. 10% of (pseudo-randomized) trials were misleading *noisy trials*, yielding to a positive reward for an incorrect association, and vice-versa. Thus, a null feedback could be produced either by a behavioral error, by a change in correct associations, or by noise. The introduction of misleading feedback prevented subjects from inferring a change in correct associations from a single unrewarded trial.

The correct set of responses to stimuli remained unchanged over a block of 36 to 54 trials. Such a block is called an *episode*. The transition from one episode to another is called an *episode switch* and was not explicitly cued. This transition imposes a change of correct responses from all stimuli, so a change of *task-set*. Task-sets were always non-overlapping from one episode to the other, that is each and every of the three stimulus-response associations differ after an episode switch. Within a given set, two stimuli were never associated with the same action.

An experimental *session* was a succession of 25 episodes for Experiment 1, and 24 episodes for Experiment 2. BOLD activity was acquired only during the 16 last episodes of Experiment 2.

In each experiment, subjects performed two distinct sessions: an open-ended and a recurrent session. In the *open-ended* session, task-sets were different in each episode, so there was no possibility for the subject to retrieve and reuse a formerly learned one. In the *recurrent* session, only 3 task-sets reoccurred across episodes (*Figure 1a*), and subjects could reuse previously learned task-sets. Subjects were not informed about the distinction between the two sessions. The order of the sessions was changed between subjects to counteract for potential session-ordering effect. Different digits were used from one session to the other.

Having to manipulate at least 3 different task-sets was crucial: indeed when there are only 2 of them, the second one could be inferred from the first one by elimination. With 3 task-sets, and after an episode switch, some exploration is required to find the next mapping to use. More possible actions than the number of stimuli were used to avoid learning the third stimulus-response association by simple elimination when two associations were already known.

## Debriefing

After each session, subjects performed a post-test debriefing. They were presented with 6 task-sets and rated them depending on their confidence in having seen them or not during the experiment. For the recurrent session, 3 out of the 6 task-sets were actually presented during the experiment. For the open-ended session, the 6 task-sets were all part of the experiment. From the debriefing of the recurrent session, subjects were classified in two different groups. *Exploiting subjects* ranked higher confidence for the 3 seen task-sets, compared to the 3 unseen task-sets. *Exploring* subjects, on the contrary, ranked at least 1 unseen task-set with more confidence than one of the 3 seen task-sets.

## Network model

The network model is based on *Rigotti et al. (2010a)*. It is composed of two interacting subnetworks, the associative and task-set networks, illustrated in (*Figure 1d*). In contrast to *Rigotti et al. (2010b)*, we do not explicitly model temporal dynamics within a trial, but instead use simplified, instantaneous dynamics between populations replacing many mixed-selective neurons (*Fusi et al., 2016*). Moreover, the feedback from the task-set network to the associative network is implemented in a simplified manner. Full details of the model implementation are given below.

## The associative network

The associative network (*AN*) is based on *Fusi et al. (2007)*. This subnetwork implements in a simplified fashion the associations between input stimuli and output actions performed by a classical

winner-take-all decision network (**Brunel and Wang, 2001**; **Wang, 2002**; **Fusi et al., 2007**; **Hertz, 2018**).

The associative network is composed of neural populations selective to a single task-related aspect, either a stimulus or an action. Each population is either active or inactive in any trial, so that the activity $\nu$ is modeled as being binary. If $S_i$ is the neural population selective to the presented stimulus, and $A_j$ is the neural population selective to the chosen action:

$$\nu(S) = \begin{cases} 1 & \text{if } S = S_i \\ 0 & \text{otherwise} \end{cases} \tag{1}$$

$$\nu(A) = \begin{cases} 1 & \text{if } A = A_j \\ 0 & \text{otherwise} \end{cases} \tag{2}$$

Any stimulus-selective neural population $\{S_i\}_{i=1..3}$ projects excitatory synapses to all response-selective neural population $\{A_j\}_{i=1..4}$. The corresponding synaptic strength, noted $J^{AN}_{S_i \to A_j}$, takes values between and 1. The behavioral output in response to a stimulus is determined based on these synaptic strengths, which moreover plastically change depending on the outcome of the trial (reward or no reward).

### Action selection in the associative network

In any given trial, following the presentation of a stimulus $S_i$, the associative network stochastically selects an action based on the strengths of the synapses from the population $\nu(S_i) = 1$ to the populations that encode actions. Specifically, the action $A_j$ is selected with the probability

$$P(A_j|S_i) = \frac{\epsilon}{n_A} + (1-\epsilon) \frac{\exp(\beta J^{AN}_{S_i \to A_j})}{\sum\limits_{k=1}^{n_A} \exp(\beta J^{AN}_{S_i \to A_k})} \tag{3}$$

where $n_A$ is the number of possible actions, $1/\beta$ is the strength of decision noise and $\epsilon$ accounts for undirected exploration, that is random lapses. The associative network therefore effectively implements a soft and noisy winner-take-all mechanism: all actions are equiprobable for high decision noise, whereas the probability of the action with the largest synaptic strength tends to 1 for low decision noise.

### Synaptic plasticity in the associative network

The learning of the basic stimulus-action associations is implemented through plastic modifications of the synaptic strengths in the associative network. Following an action, the synaptic strengths are updated according to a reward-modulated, activity-dependent learning rule:

$$J^{AN}_{S_i \to A_j} \leftarrow J^{AN}_{S_i \to A_j} + \alpha_+(r, \nu(S_i), \nu(A_j)) \cdot (1 - J^{AN}_{S_i \to A_j}) - \alpha_-(r, \nu(S_i), \nu(A_j)) \cdot J^{AN}_{S_i \to A_j} \tag{4}$$

where $r$ is the obtained reward ($r = 0$ or 1), and $\alpha_+$ and $\alpha_-$ are respectively the rates of potentiation and depression which depend on the reward as well as the activity of pre- and post-synaptic populations. Note that the update rule implements soft bounds on synaptic strengths, and ensures biological plausible saturation of neural activity, as well as forgetfulness (**Amit and Fusi, 1994**; **Fusi, 2002**; **Fusi et al., 2007**; **Ostojic and Fusi, 2013**).

The synaptic plasticity is local, so that only synapses corresponding to the active pre-synaptic population $S_i$ are updated ($\nu(S_i) = 1$). Moreover, for simplicity, all non-zero potentiation and depression rates are equal and given by a parameter $\alpha$. We therefore have

$$\alpha_+(r = 1, \nu(S_i) = 1, \nu(A_j) = 1) = \alpha_-(r = 1, \nu(S_i) = 1, \nu(A_j) = 0) = \alpha \tag{5}$$

if the reward is positive, and

$$\alpha_+(r = 0, \nu(S_i) = 1, \nu(A_j) = 0) = \alpha_-(r = 0, \nu(S_i) = 1, \nu(A_j) = 1) = \alpha \tag{6}$$

if the reward is null. All other potentiation and depression rates are zero.

The simplest implementation of the associative network therefore has three free parameters: the learning rate $\alpha$ and noise parameters $\beta$ and $\epsilon$. When fitting the model to human behavior, we have examined the possibility of adding complexity by distinguishing the learning rates corresponding to distinct reward and pre/post- synaptic events. The presented results on model fits, model dynamics and model predictions concerning the recurrent session are not modified by this extension.

## The task-set network

The task-set network (*TN*) is composed of mixed-selective neural populations, which are selective to conjunctions $S_iA_j$ of one stimulus and one action. As in the associative network, the activity $\nu$ of each population in the task-set network is represented as binary (either active or inactive, $\nu(S_iA_j) \in \{0, 1\}$). The task-set network is fully connected: any neural population $S_iA_j$ projects excitatory synapses to all other neural populations $S_kA_l$, with strength $J^{TN}_{S_iA_j \rightarrow S_kA_l} \in [0, 1]$. The strengths of these synapses are plastically updated, and determine the co-activation of populations in the task-set network. This co-activation effectively encodes task-sets. Full details of the model implementation are given below.

### Activation of populations in the task-set network

At each trial, a stimulus $S_i$ is presented and the associative network selects an action $A_j$. In the task-set network, this leads to the activation of the population $S_iA_j$ :

$$\nu(S_iA_j) = 1 \tag{7}$$

Depending on the synaptic strengths, this may in turn lead to the co-activation of other populations in the task-set network. Specifically, if the synaptic strength $J^{TN}_{S_iA_j \rightarrow S_kA_l}$ is greater than the parameter $g_I$, the population $S_kA_l$ is immediately activated.

For all $(S_k, A_l)$ with ($k \neq i$ or $l \neq j$) :

$$\nu(S_kA_l) = \begin{cases} 1 & \text{if } J^{TN}_{S_iA_j \rightarrow S_kA_l} \geq g_I \\ 0 & \text{otherwise} \end{cases} \tag{8}$$

This step is iterated until no additional population gets activated. Here the parameter $g_I$ represents an inhibitory threshold equivalent to a constant negative coupling between all populations in the task-set network, and implements in a simplified way a competition between excitatory neural populations through recurrent feedback inhibition.

These activation dynamics are assumed to be fast on the timescale of a trial, and therefore implemented as instantaneous.

### Synaptic plasticity in the task-set network

The synapses in the task-set network are updated following an unsupervised, Hebbian plasticity rule. This update is driven by the sequential activation of neural populations in the task-set network, and thus by the associative network dynamics (*Figure 1d*). When two populations in the task-set network are activated on two consecutive trials, the synapses connecting them are potentiated. Noting $S_tA_t$ the task-set network neural population at trial $t$, and $S_{t+1}A_{t+1}$ the neural population at trial $t+1$, this potentiation is given by

$$J^{TN}_{S_tA_t \rightarrow S_{t+1}A_{t+1}} \leftarrow J^{TN}_{S_tA_t \rightarrow S_{t+1}A_{t+1}} + Q_P \cdot (1 - J^{TN}_{S_tA_t \rightarrow S_{t+1}A_{t+1}}) \cdot \nu(S_tA_t) \cdot \nu(S_{t+1}A_{t+1}) \tag{9}$$

where the parameter $Q_P$ represents the learning rate for potentiation.

Moreover, at each trial, all the synapses from the active neural population $\nu(S_tA_t) = 1$ are depressed (*pre-activated depression* [**Ostojic and Fusi, 2013**]), implementing an effective homeostatic control. This depression is given by

$$J^{TN}_{S_tA_t \rightarrow S_kA_l} \leftarrow J^{TN}_{S_tA_t \rightarrow S_kA_l} - Q_M \cdot J^{TN}_{S_tA_t \rightarrow S_kA_l} \cdot \nu(S_tA_t) \tag{10}$$

where $Q_M$ is the rate of depression.

The ratio $Q_P/Q_M$ between the potentiation and the depression rates determines the asymptotic values of the synaptic strengths (*Ostojic and Fusi, 2013*). To produce co-activation of populations in the task-set network and therefore the learning of task-sets, this asymptotic value needs to be higher

than the inhibition threshold $g_I$. To avoid any redundancy between the parameters we fixed $g_I$ to 0.5 and $Q_M = Q_P/10$, so that $Q_P$ is the only free parameter. Using different sets of values for $g_I$ and $Q_P/Q_M$ did not modify the fits and the results as long as $g_I$ was large enough to filter out the fluctuations of non-task-set synaptic weights (*Figure 5—figure supplement 2*).

## Interaction between associative network and task-set network

To implement the effect of learning in the task-set network on the output of the model, the pattern of activity in the task-set network needs to influence the activation in the associative network. In our model this interaction is implemented in a simplified fashion.

If the current trial was rewarded, the activation in the associative network on the next trial is biased towards the stimulus-action combinations that correspond to activated populations in the task-set network. This effective *inference signal* is implemented by modulating the strength of synapses in the associative network. Specifically, if $r = 1$ and $\nu(S_k A_l) = 1$ in the task-set network on the previous trial, the synaptic strength from $S_k$ to $A_l$ in the associative network is increased.

$$J_{S_k \to A_l}^{AN} \leftarrow J_{S_k \to A_l}^{AN} + J_{INC} \cdot (1 - J_{S_k \to A_l}^{AN}) \cdot \nu(S_k A_l) \tag{11}$$

where the parameter $J_{INC}$ specifies the strength of the inference signal.

The strength of the inference signal thus corresponds to the discrepancy between the task-set network prediction and the associative network encoding when a reward is received (or its negative counterpart, the compatibility).

## Model fitting to behavioral data and simulation

The model without task-set inference has three free parameters, and the model with task-set inference has five free parameters (taking into account that we fixed the parameters $g_I$ and $Q_P/Q_M$ in the task-set network). The parameter set is composed of the associative network learning rate $\alpha$, the task-set network learning rate $Q_P$, the parameters of the soft and noisy winner-take-all mechanism (decision noise $1/\beta$ and uncertainty $\epsilon$), and the inference strength $J_{INC}$ from task-set network to associative network connectivity. Both models were fitted to behavioral data using the standard maximum likelihood estimation (MLE). We provide the model with the subject's set of actions and we define the model *performance on a trial* as the model's likelihood of observing the subject' response at the next trial. The Bayesian information criterion (*Figure 5a* and *Figure 5—figure supplement 1*) uses the likelihood computed by MLE but introduces a penalty term for model complexity depending on the size of the observed sample. Following Figure 2 of *Collins and Koechlin (2012)*, the BIC was computed as the opposite of equation (4.139) of *Bishop (2007)*. We also compared the AIC (Akaike information criterion) for both models and reached identical conclusions (see *Figure 5—source data 1*). A larger log-likelihood and lower BIC and AIC correspond to best model fits. We combined a grid search on initial parameters values with a gradient descent algorithm from the SciPy optimization toolbox. Parameters were estimated subject by subject.

On average over subjects (see *Figure 5—source data 1*), the learning rate in the associative network ($\overline{\alpha} = 0.35$, $\sigma = 0.0073$) is twice the learning rate in the task-set network ($\overline{Q_P} = 0.17$, $\sigma = 0.0070$). The inference strength from the task-set network to the associative network is high in the recurrent session ($\overline{J_{INC}} = 0.70$, $\sigma = 0.037$), and its value is significantly lower in the open-ended session ($\overline{J_{INC}} = 0.16$, $\sigma = 0.0062$, see *Figure 5b*).

We also compared model simulations ex post (model recovery *Palminteri et al., 2017*), with and without task-set inference. In a simulation, the model's actions are random depending on the trial by trial probability set computed from the winner-take-all mechanism. The *model performance* is now the probability predicted by the model for the correct action, at each trial. Model simulation reproduced model fits and data, which ensured that we are not overfitting subjects' data.

## fMRI whole brain analysis

The model-based fMRI analysis was performed with SPM 12. The detail of the data acquisition can be found in *Donoso et al. (2014)*.

All parametric modulators were z-scored to ensure between regressor and between subjects commensurability (*Lebreton et al., 2019*; see also *Figure 7—figure supplement 2*). For each onset, they were orthogonalized to avoid taking into account their shared variance. fMRI data were

analyzed using an event-related general linear model. Regressors of non-interest include subjects' lapses, post-pause trials at the beginning of each scanning run, and movement parameters. Event-related regressors of interest modeled separately the onset decision (stimulus presentation timing, covering the decision time window) and the onset feedback (outcome presentation timing). The regressors were based on the best fitting parameters at the subject level.

At the onset decision, the regressor included orthogonalized parametric modulations following this order:

- The first modulator was the time-series of reaction times, an index of trial difficulty and a specific motor preparation-related activity.
- The second modulator was the associative network synaptic strength from the presented stimulus selective neural population to the chosen action selective neural population. We call this parameter $W_{chosen}$ and it is also an index of trial difficulty.

At the onset feedback, the regressor included orthogonalized parametric modulations following this order:

- The first parametric modulator was $W_{chosen}$. This control ensures that the correlations observed are not simply caused by the monitoring of the certainty on the chosen association (prediction error) or else the trial difficulty.
- The second parametric modulator was the time series of positive rewards.
- The third parametric modulator was the trial-by-trial average value of the inference signal from the task-set network to the associative network. It was thus the average, over the number of connections implicated, of the task-set network inference on the update of associative network synaptic weights. We call it *TN inference*.

All the mentioned time series were convolved with the hemodynamic response function to account for the hemodynamic lag effect.

The subject by subject statistical maps were combined to make generalizable inferences about the population. We used a *random effect analysis* approach (*Holmes and Friston, 1998*). We identified activations using a significance threshold set to $p = 0.05$ (familywise error FWE corrected for multiple comparison over the whole brain).

For conciseness, and because mixed-selectivity has been found in prefrontal cortex (*Fusi et al., 2016*), we do not report posterior activations (parietal, temporal and occipital lobes).

Note that we did a preliminary control analysis using the link between the associative network and Q-learning (*Watkins and Dayan, 1992*) by searching for any correlation between BOLD activity and the prediction error, that is the difference between the perceived outcome and the associative network synaptic strength of the trial-by-trial chosen association. As expected from previous studies (*Tanaka et al., 2004*; *O'Doherty et al., 2004*; *Daw et al., 2006*; *Kim et al., 2006*; *Lebreton et al., 2009*), we found ventromedial prefrontal cortex and striatal activity to correlate positively in the recurrent and in the open-ended session. The MNI peak coordinates and number of voxels in the cluster were respectively $[-12, 56, 20]$, $T = 11.2$, 901 voxels in the recurrent session, and $[-12, 8, -12]$, $T = 14.3$ and 1419 voxels in the open-ended session.

We investigated neural correlates of the trial-by-trial synaptic strength of chosen association in the associative network, at the onset decision ($W_{chosen}$, controlling for trial difficulty, through time-series of reaction times). Results are shown in *Figure 7—source data 1*. We found positive correlations in striatum and ventromedial prefrontal cortex, and negative correlations in dorsal anterior cingulate cortex, anterior supplementary motor area and lateral prefrontal cortex, in both experimental sessions. These results are consistent with previous findings in the field of value-based decision making (*Tanaka et al., 2004*; *Daw et al., 2006*; *Lebreton et al., 2009*; *Chib et al., 2009*; *Alexander and Brown, 2011*; *Donoso et al., 2014*; *Neubert et al., 2015*; *Palminteri et al., 2015*).

## One-way ANOVA and second control ROI analysis

In order to test the hypothesis of a specific effect of task-set retrieval, we extracted the betas from medial and lateral prefrontal nodes, and compared them from the two conditions: the recurrent and the open-ended session. This comparison was valid as soon as the region of interest is selected independently from the statistical maps of betas (*Poldrack, 2007*), that is the selected ROI need to be based on a different contrast than the one currently studied. We defined a functional network by the co-activations in both sessions, for the trial-by-trial task-set network inference signal to the

associative network (ANOVA REC+OE for dorsomedial and dorsolateral prefrontal cortex, ANOVA -REC-OE for ventromedial prefrontal cortex, FWE 0.05, *Table 1*), which thus did not promote differences. Our ROI of ventromedial, dorsomedial and dorsolateral prefrontal cortex were selected from the obtained thresholded maps (FWE $p = 0.05$) from this ANOVA analysis, and were used to test differences between REC and OE (in *Figure 7a*).

We also looked at the correlations between magnetic resonances responses at feedback onset, in these ROI, to a similar general linear model where the inference signal had been replaced by five sparser time-series constituted of only one trial per episode and the corresponding events shifted one or two trials preceding and following it. This trial was chosen as the first chunked trial of the model behavioral predictions (per episode, if it existed from sufficient learning, *Figure 6a,b,c*). Results are displayed in *Figure 7b*.

We further controlled our results by running other independent ROI analysis using:

- the Stanford Functional Imaging in Neuropsychiatric Disorders Lab (*Shirer et al., 2012*)
- the Glasser parcellation (*Glasser et al., 2016*)
- the Neurosynth meta-analysis (*Yarkoni et al., 2011*)
- the WFU PickAtlas (*Lancaster et al., 1997*; *Lancaster et al., 2000*; *Maldjian et al., 2003*)

Results are shown in *Figure 7—source data 2*.

## Neural correlates of the inference signal in the hippocampus

We used an ROI of hippocampus from the WFU PickAtlas (*Lancaster et al., 2000*). Using an ROI from the Glasser parcellation (*Glasser et al., 2016*) or the Neurosynth meta-analysis (*Yarkoni et al., 2011*) provided comparable results. Only regressors of the synaptic strength of the chosen association at the onset decision, and of positive rewards at the onset feedback, showed positive correlations in both sessions, but the difference between sessions was not significant (*Figure 7—figure supplement 1*). This is consistent with the general role of the hippocampus in associative binding (*Davachi and DuBrow, 2015*). In contrast, there was no significant activation of hippocampus corresponding to the task-set network inference to the associative network, neither in the recurrent nor in the open-ended session. Note that we are regressing the value of the inference strength from the task-set network biasing the decision network while learning occurs, as a cognitive control signal would do. It's an indirect measure of memory encoding in the task-set network, instead of the actual value of synaptic weights.

## Software

All simulations were done with Python 2.7 (using numpy and scipy, and the scikit-learn package; *Pedregosa et al., 2011*). The fMRI analysis was done with Matlab and SPM12 (*Ashburner et al., 2014*). Code is available at https://github.com/florapython/TemporalChunkingTaskSets (*Bouchacourt, 2020*; copy archived at https://github.com/elifesciences-publications/TemporalChunkingTaskSets).

## Acknowledgements

FB was funded by Ecole des Neurosciences de Paris Ile-de-France and Region Ile de France (DIM Cerveau and Pensee). SP is supported by an ATIP-Avenir grant (R16069JS) Collaborative Research in Computational Neuroscience ANR-NSF grant (ANR-16-NEUC-0004), the Programme Emergence(s) de la Ville de Paris, the Fondation Fyssen and the Fondation Schlumberger pour l'Education et la Recherche. SO is funded by the Ecole des Neurosciences de Paris Ile-de-France, the Programme Emergences of City of Paris, Agence Nationale de la Recherche grants ANR-16-CE37- 0016–01 and ANR-17-ERC2-0005-01. This work was supported by the program 'Investissements d'Avenir' launched by the French Government and implemented by the ANR, with the references ANR-10-LABX-0087 IEC and ANR11-IDEX-0001–02 PSL Research University.

## Additional information

### Funding

| Funder | Grant reference number | Author |
|---|---|---|
| Ecole des Neurosciences de Paris Ile-de-France | Doctoral Fellowship | Flora Bouchacourt |
| Agence Nationale de la Recherche | ANR-16-CE37- 0016-01 | Srdjan Ostojic |
| Agence Nationale de la Recherche | ANR-17-ERC2-0005-01 | Srdjan Ostojic |
| Inserm | R16069JS | Stefano Palminteri |
| Agence Nationale de la Recherche | ANR-16-NEUC-0004 | Stefano Palminteri |
| Fondation Fyssen | | Stefano Palminteri |
| Schlumberger Foundation | | Stefano Palminteri |
| Ecole des Neurosciences de Paris Ile-de-France | | Srdjan Ostojic |
| Region Ile de France (DIM Cerveau et Pensee) | | Flora Bouchacourt |

The funders had no role in study design, data collection and interpretation, or the decision to submit the work for publication.

### Author contributions

Flora Bouchacourt, Conceptualization, Formal analysis, Validation, Investigation, Visualization, Methodology; Stefano Palminteri, Validation, Visualization, Methodology; Etienne Koechlin, Data curation, Supervision, Methodology; Srdjan Ostojic, Conceptualization, Supervision, Methodology, Project administration

### Author ORCIDs

Flora Bouchacourt 🆔 https://orcid.org/0000-0002-8893-0143
Stefano Palminteri 🆔 http://orcid.org/0000-0001-5768-6646
Srdjan Ostojic 🆔 https://orcid.org/0000-0002-7473-1223

### Ethics

Human subjects: Participants provided written informed consent approved by the French National Ethics Committee.

### Decision letter and Author response

Decision letter https://doi.org/10.7554/eLife.50469.sa1
Author response https://doi.org/10.7554/eLife.50469.sa2

## Additional files

### Supplementary files

• Transparent reporting form

### Data availability

Code has been uploaded to https://github.com/florapython/TemporalChunkingTaskSets (copy archived at https://github.com/elifesciences-publications/TemporalChunkingTaskSets). Statistical maps corresponding to human subjects data have been uploaded to Neurovault (https://neurovault.org/collections/6754/).

The following datasets were generated:

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
