## [Decision Letter]

**Acceptance summary:**

This paper combines a computational model with analysis of human data to explain how humans learn to use temporal proximity to learn task sets and switch between the task sets.

**Decision letter after peer review:**

Thank you for submitting your article "Temporal chunking as a mechanism for unsupervised learning of task-sets" for consideration by *eLife*. Your article has been reviewed by three peer reviewers, one of whom is a member of our Board of Reviewing Editors, and the evaluation has been overseen by Barbara Shinn-Cunningham as the Senior Editor. The reviewers have opted to remain anonymous.

The reviewers have discussed the reviews with one another and the Reviewing Editor has drafted this decision to help you prepare a revised submission.

Summary:

This paper develops a model of task switching that uses synaptic plasticity that includes temporal contiguity to chunk the tasks, and compares the model to human behaviour and fMRI findings. The combination of all these 3 ingredients is interesting and allows for a precise comparison between model and data.

Essential revisions:

1) In the model the weights in the Association Network are gradually updated. One would imagine that once participants have been exposed to a limited number task sets in the recurrent task, they can switch very rapidly (and perhaps without invoking synaptic plasticity). I.e. an alternative to the model is that the connections in the Association Network are directly gated by the task network. I would like to see the predictions of the model regarding this, namely I would like to know how the data plotted in Figure 6B evolve as learning progresses, and compare data to model.

2) In the Task Network starting from small weights there is a sudden emergence of a cluster when weights cross g_I (as the Discussion section mentions it should actually be modelled as an attractor in a Hopfield network). Is there evidence for such a switch in the data?

3) Related to that, it would be interesting to see the model prediction for the case that a new task-set partly overlaps with a learned one and one needs to un-learn the previous task-set.

4) While the full model fits better than the non-chunking model in terms of AIC, I would really like to see the actual fit of both models.

5) Clarification of model. I found the description of the model to be unclear and more can be done to explain why certain modeling choices were made. Specifically, was this an attractor network with internal dynamics and did the activity in the network persist from trial-to-trial? Equations 5 and 7 suggest that learning occurs in the difference between trial-to-trial activity in the task-set network and that the activity in the task-set network on one trial influence the weights of the associative network in the subsequent trial. I found this surprising – based on the Introduction, I was expecting to see a recurrent model in which activity was sustained across trials.

6) It appears the model was fit separately for the "recurrent" and "open-ended" session – why was this done? I find it implausible that subjects would have two sets of parameters that best describe their behavior in this task (as this analysis implies, especially Figure 5B), especially as the two sessions were un-signaled. It makes more sense to me to fit a single model to all of the data and examine the trial-by-trial log-loss. Does the model have the expressiveness to capture both sessions under the same parameter set? How does this change the analysis?

7) The comparison to previous computational models is lacking. There is an existing neuro-biological model of task-set learning with physiological constraints (Collins and Frank, 2013) that warrants comparison, especially as the first author of that paper was the first author of the paper re-analyzed here. The paper itself was cited but described as "without any physiological constraint" which is not a fair description of the work. More broadly, the comparison to basal ganglia/PFC gating models is worth making (e.g. Frank and Badre, 2012; Rougier et al., 2005), as it is not obvious to me how the author's model makes different predictions from prior biological models. I would also note that there are algorithmic similarities between this model, the successor representation (Russek et al., 2017) and the Temporal context model (Gershman et al., 2012 explains this most clearly).

8) Were there any findings in the hippocampus related to the task-set network? There is a long literature of sequential dependencies being represented in the hippocampus (see Davachi and Dubrow, 2015 for review), and it would not be surprising if the model accounted for hippocampal activity as well.

9) Conceptually, the model is a minor extension of the existing model of Rigotti et al., (2010a,b) from the Fusi group with application to the human behavioral task at hand. It combines standard reinforcement learning of stimulus-response associations with a context-network that encodes the current task (called task-set network). To generalize across the three different possible stimuli in a given task the context network uses a Hebbian unsupervised learning rule. The most problematic part, from a biological perspective, is the fact that the feedback from the context network to the stimulus-response association network acts directly on the connections inside the stimulus-response association network (as opposed to input to neurons which would be the standard way of doing it). Unless this is justified by additional information (could be text, links to biological literature of biological synapses, or simulations of a more complex, but neuron-based model) I consider this as a potential flaw of the model. The link to the Fusi paper is not sufficient in my opinion.

10) Motivation of model. Even though the authors imply that they have put together a biologically plausible model, it actually looks more like a very high-level abstract model where a whole population of neurons is replaced by a single binary switch; where there are exactly as many binary switches as there are potential combinations of stimuli and responses; and where learning rules are written as algorithmic updates rather than as weight changes driven by pre-and postsynaptic activity.

Thus, I was not convinced about the approach until I understood in Figure 6 and the associated text that the authors actually use this abstract model (which only has a few free parameters, number to be clarified, see below) to fit behavioral data on a subject-by-subject basis. For me this is the strongest point of the paper which eventually makes me support publication after some modifications.

Nevertheless, it is unclear to me whether the same learning rules would also work if the neural network model included a large number of randomly connected neurons, heterogeneity, and continuous-values rate units. I understand that the competitive dynamics would make the continuous-values units near-binary after learning, but the dynamics during learning might still be different, and hence convergence times as well (which could influence the results of the paper). Alternatively, reformulate the text to make it clear that the modeling work uses a rather abstract model with abstract learning rules.

---

## [Author Response]

Essential revisions:1) In the model the weights in the Association Network are gradually updated. One would imagine that once participants have been exposed to a limited number task sets in the recurrent task, they can switch very rapidly (and perhaps without invoking synaptic plasticity). I.e. an alternative to the model is that the connections in the Association Network are directly gated by the task network. I would like to see the predictions of the model regarding this, namely I would like to know how the data plotted in Figure 6B evolve as learning progresses, and compare data to model.

We thank the reviewers for this question. In our model, the weights in the Association Network are modified by a combination of gradual updates and direct binary gating from the task-set network. The relative contributions of these two types of updates are specified by the parameter J_INC_: if J_INC_=0 updates are only gradual; if J_INC_=1 gating from the task-set network directly switches the weights to their maximal value. This parameter J_INC_ is fitted on the behavioral data subject-by-subject (Figure 5B), and the obtained values indicate that the behavior is best described by a combination of gradual updates and direct gating.

To provide more evidence and intuition for this point, we now show how the data of Figure 6B evolve as learning progresses, both within an episode (averaged over episodes), and between episodes.

First, we plotted the same data as in Figure 6B but by adding trials before and after the first rewarded trial. Results are displayed in Figure 5C and show that the switch in behavior is not instantaneous, but instead a combination of an instantaneous jump and more gradual adjustments.

Second, to show the evolution over the whole session, in Figure 6—figure supplement 2A we split data of Figure 6B as a function of episode number. This figure shows that even at the end of the session, the retrieval of a task-set is not complete and instantaneous, so that a mixture of gating and gradual update is present.

The need for this combination of gradual and sudden updates can be understood from the necessity to filter out misleadingly rewarded noisy trials (Figure 6E). If the weight update was controlled solely by a gating-induced switch, the network would be unable to perform a correct association at the trial following a misleading reward. We plotted in Figure 6—figure supplement 2B the data of Figure 6E as a function of episode number, i.e. as learning evolves. This figure shows that the subjects’ probability of making a correct association after a misleadingly rewarded trial is not null, even for the last episodes, after extensive learning of the three task-sets.

To clarify these points in the text, we have performed the following changes:

– When introducing the model, we added (subsection “A network model for learning task-sets by chunking stimulus-response pairs”): “The synaptic weights in the AN are therefore modified by a combination of sudden changes due to the inference signal and more gradual updates. The relative contribution of these two mechanisms is determined by a parameter that represents the strength of task-set retrieval (if it is zero, there is no retrieval).”

– When describing the fits to the data, we added (subsection “Fitting the model to behavioral data”): “In particular, the full model captured well the behavior at episode change, where following a first correct trial the subjects’ performance exhibited a sudden increase corresponding to task-set retrieval, combined with more gradual changes (Figure 5C).”

2) In the Task Network starting from small weights there is a sudden emergence of a cluster when weights cross g_I (as the Discussion section mentions it should actually be modelled as an attractor in a Hopfield network). Is there evidence for such a switch in the data?

We thank the reviewer for this question. In Figure 6, “independent” and “chunked” refer to trials in which the corresponding cluster is respectively absent or present. The analyses in Figure 6B,C,E therefore provide evidence for a difference in behavior depending on whether a cluster is present or not in the task-set network. Figure 8B provides evidence for a difference in neural activity depending on the presence of a cluster.

In these analyses, all episodes respectively with or without a cluster were grouped together. To address the reviewer’s question, we examined the evidence for the effect on the behavior of a sudden emergence of a cluster. In the new Figure 6—figure supplement 2A, we split the data of Figure 6B as a function of episode number since the appearance of a cluster in the model. The resulting data are noisy, as each data point now contains only one trial per subject but are consistent with a sudden change of behavioral performance when a cluster emerges.

Note that in the model, a cluster corresponding to a task-set emerges progressively depending on the temporal contiguity of associations. First, two stimulus-action associations are chunked together, then the third one is eventually added to the cluster. We now make this progressive process clearer in Figure 3—figure supplement 1. In Figure 6 and Figure 6—figure supplement 2, we identify the first trial where (at least) two associations are chunked together in the model.

To clarify these points in the text, we have performed the following changes:

– In subsection “Testing model predictions for task-set retrieval” we added: “The model additionally predicted a sudden switch in behavior after two stimulus-response associations were chunked together. Splitting the data of Figure 6 as a function of episode number revealed a pattern consistent with such a switch (Figure 6—figure supplement 2).”

– In subsection “Speed-accuracy trade-off for learning task-sets in the network model” we added: “First, two stimulus-action associations are chunked together, then the third one is eventually added to the emerging cluster (Figure 3—figure supplement 1).”

3) Related to that, it would be interesting to see the model prediction for the case that a new task-set partly overlaps with a learned one and one needs to un-learn the previous task-set.

We thank the reviewer for this suggestion. In the revised manuscript, we studied two cases: the case where a newly introduced task-set partly overlaps with a previously learned one, and the case where two task-sets partially overlap from the beginning of learning. In both cases, our model predicts that the overlapping association induces a decrease in performance on the following trial. In the first case, this decrease is transient, while in the second case it is permanent. These predictions are now reported in a new subsection “Predictions for experiments with overlapping task-sets”, and we detail them below.

For the first case (Figure 3—figure supplement 2A-E), we simulated the recurrent session till episode 25, as in Figure 3 A,C,E,G. After episode 25, we introduced a fourth task-set (task-set 4, see Figure 3—figure supplement 2A) that had one overlapping stimulus-action association with task-set 1 (association [5j]). When this new task-set is introduced, in the task-set network the overlapping association [5j] is chunked with stimulus-response associations corresponding to task-set 1. In consequence, any trial on which stimulus 5 is shown leads to an incorrect retrieval of task-set 1, and therefore errors on the next trial (Figure 3-figure supplement 2Ee). This is analogous to maladaptive retrieval examined in Figure 6D,E. Synaptic depression in the task-set network eventually breaks away the association [5j] from the cluster corresponding to task-set 1, and chunks it with task-set 4, at which point the performance on the trials following stimulus 5 increases (Figure 3—figure supplement 2B).

For the second case, we considered (in Figure 3—figure supplement 2F,G) a session with three task-sets, among which two shared the same stimulus-response association (task-set 1 and task-set 3 shared the association [5j], see Figure 3—figure supplement 2F). In the model, the shared association becomes either chunked with both task-sets, or with neither of the two. In either case, it is uninformative on the correct task-set. The model therefore predicts a decrease of performance on the trial following the shared association, in particular at the beginning of an episode (Figure 3—figure supplement 2G).

These predictions could be tested in future work.

4) While the full model fits better than the non-chunking model in terms of AIC, I would really like to see the actual fit of both models.

The model was fitted with maximum likelihood estimation on the sequence of actions (see Materials and methods section), so the fit cannot be directly visualized. Nevertheless, we added in Figure 5—figure supplement 2B the comparison between the two models (the chunking model ANTN, and the non-chunking model AN) by plotting the performance after the episode switch, while in Figure 5C we added the performance following the first correct trial. These curves were not directly fitted, yet they display the key difference between the chunking and the non-chunking model: the chunking model captures well the retrieval of a task-set following a first correct trial (sudden jump in performance), while the non-chunking model does not. Note that the likelihood in the MLE fit weighs equally the trials after an episode switch and trials at the end of the episode, when the behavior is stable.

In addition to this, Figure 5C shows the subject-by-subject difference in BIC scores for the two models in the recurrent session, for Experiment 1. Figure 5—figure supplement 2a shows the corresponding data for Experiment 2. We added the likelihood itself in Figure 5—figure supplement 2C,D. Subject-averaged AIC/BIC scores were also reported in Figure 5—figure supplement 1.

To clarify these points in the main text, we have performed the following change:

In “Fitting the model to behavioral data” we added (subsection “A network model for learning task-sets by chunking stimulus-response pairs”): “In particular, the full model captured well the behavior at episode change, where following a first correct trial the subjects’ performance exhibited a sudden increase corresponding to task-set retrieval, combined with more gradual changes (Figure 5C).”

5) Clarification of model. I found the description of the model to be unclear and more can be done to explain why certain modeling choices were made. Specifically, was this an attractor network with internal dynamics and did the activity in the network persist from trial-to-trial? Equations 5 and 7 suggest that learning occurs in the difference between trial-to-trial activity in the task-set network and that the activity in the task-set network on one trial influence the weights of the associative network in the subsequent trial. I found this surprising – based on the Introduction, I was expecting to see a recurrent model in which activity was sustained across trials.

We apologize for this confusion. We did not use an attractor network with internal dynamics, but a simplified model as we focused on chunking mechanisms and on the effect of temporal order of events. Indeed, the reviewer is right, learning occurs in the difference between trial-to-trial activity in the task-set network, and the activity in the task-set network on one trial indeed influence the weights of the associative network in the subsequent trial for the next decision to be made.

We have now clarified this in the Introduction, Discussion section and Materials and methods section.

6) It appears the model was fit separately for the "recurrent" and "open-ended" session – why was this done? I find it implausible that subjects would have two sets of parameters that best describe their behavior in this task (as this analysis implies, especially Figure 5B), especially as the two sessions were un-signaled. It makes more sense to me to fit a single model to all of the data and examine the trial-by-trial log-loss. Does the model have the expressiveness to capture both sessions under the same parameter set? How does this change the analysis?

The new Figure 5—figure supplement 4 shows the comparison between fitting together and separately the recurrent and open-ended sessions. As expected, when fitting the two sessions together, the model finds a middle ground. The BIC scores are significantly worse for each session, although the magnitude of the difference between fitting separately or together is not very large (Figure 5—figure supplement 4A). Crucially, when fitting the two sessions together, the fitted inference strength J_INC_ is much lower than when the recurrent session is fitted separately (Figure 5—figure supplement 4B), so that the model is not able to capture the task-set retrieval after the first correct trial (Figure 5—figure supplement 4C). Hence, although the changes in BIC scores are small, when fitted to the two sessions together the model does not capture the main effect of interest.

The model therefore does not have the expressiveness to capture both sessions under the same parameter set. This is due to the fact that the model contains only five free parameters, and these parameters need to be modified to capture changes in the task statistics. It is well-known that monkeys, rats, and humans adopts different learning rates in similar tasks when the event statistics change (see e.g. Behrens et al., 2007). Similarly, in our model a change in the learning rate Q_P_ of the task-set network is needed to capture both the recurrent and open-ended sessions (see fitted parameter values in Figure 5—figure supplement 1). A more complex model could dynamically adapt this learning rate, at the expense of additional parameters.

To clarify these points in the text, we have performed the following changes:

– In subsection “Fitting the model to behavioral data” we added: “Note that the models were fitted separately on the recurrent and open-ended sessions. Fitting the two sessions together led to a small but significant degradation of fitting performance (Figure 5—figure supplement 4A). More importantly, the models fitted simultaneously on both sessions were not able to capture the sudden increase in performance revealing task-set retrieval after the first correct trial (Figure 5—figure supplement 4C). This failure can be traced back to the need to adapt the learning rate in the task-set network between the two sessions, as previously observed for changes in task statistics (Behrens, 2007). Indeed, when the two sessions are fitted separately, the values of this learning rate strongly differ. In the recurrent session, on average over subjects, the learning rate in the task-set network (Q_P_ = 0.17, σ = 0.0070) is half of the learning rate in the associative network (α = 0.35, σ = 0.0073), which is consistent with our initial prediction that the learning rate in the task-set network needs to be slower than in the associative network.”

7) The comparison to previous computational models is lacking. There is an existing neuro-biological model of task-set learning with physiological constraints (Collins and Frank, 2013) that warrants comparison, especially as the first author of that paper was the first author of the paper re-analyzed here. The paper itself was cited but described as "without any physiological constraint" which is not a fair description of the work. More broadly, the comparison to basal ganglia/PFC gating models is worth making (e.g. Frank and Badre, 2012; Rougier et al., 2005), as it is not obvious to me how the author's model makes different predictions from prior biological models. I would also note that there are algorithmic similarities between this model, the successor representation (Russek et al., 2017) and the Temporal context model (Gershman et al., 2012 explains this most clearly).

We thank the reviewer for pointing out these omissions. We have now modified the Introduction to clarify that both our work and the works cited above consider models at an intermediate level of description between biology and behavior. In the Discussion section, we have added a paragraph comparing our approach with basal-ganglia/PFC models, and we noted the link with the successor representation and the Temporal Context model (Introduction and subsection “Temporal chunking as a mechanism for hierarchical learning, multi-step transition maps and generalization.”).

In particular, we added the following paragraph (Discussion section):

– “Previous computational models of task-set-based behavior fall into two broad categories (Frank and Badre, 2011). […] This model was moreover developed to test a specific hypothesis, namely that task-set learning relies on Hebbian chunking of individual stimulus-response associations.”

8) Were there any findings in the hippocampus related to the task-set network? There is a long literature of sequential dependencies being represented in the hippocampus (see Davachi and Dubrow, 2015 for review), and it would not be surprising if the model accounted for hippocampal activity as well.

We thank the reviewer for suggesting the possible correlations in the hippocampus related to the task-set network inference to the associative network. Results are displayed in Figure 8—figure supplement 3 and reported in the Materials and methods section. We used an ROI of hippocampus from the WFU PickAtlas (Lancaster et al., 2000). Using an ROI from the Glasser parcellation (Glasser et al., 2016) or the Neurosynth meta-analysis (Yarkoni et al., 2011) provided comparable results. Only regressors of the synaptic strength of the chosen association at the onset decision, and of positive rewards at the onset feedback, showed positive correlations in both sessions, but the difference between sessions was not significant. This is consistent with the general role of the hippocampus in associative binding (Davachi and Dubrow, 2015).

In contrast, there was no significant activation of hippocampus corresponding to the task-set network inference to the associative network, neither in the recurrent nor in the open-ended session. Note that we are regressing the value of the inference strength from the task-set network biasing the decision network while learning occurs, as a cognitive control signal would do. It’s an indirect measure of memory encoding in the task-set network, instead of the actual value of synaptic weights. In the sequence learning experiments mentioned in (Davachi and Dubrow, 2015), a measure of successful memory encoding in neural activity is compared to a recognition test performance post-hoc, either using BOLD response (e.g. Hales and Brewer, 2010, Qin et al., 2007, Kumaran et al., 2006), or a measure of correlation between hippocampal neural activity (e.g. Paz et al., 2010). Other mentioned studies use segmented ROIs, higher anatomical resolution, and multivariate pattern analysis to study the similarity between random patterns presentation before and after the sequence exposure to avoid the confound of temporal autocorrelation in BOLD responses due to the creation of similar representations (e.g. Shapiro et al., 2012, but see also Schendan et al., 2003 using alternation of random and sequence blocks).

9) Conceptually, the model is a minor extension of the existing model of Rigotti et al., (2010a,b) from the Fusi group with application to the human behavioral task at hand. It combines standard reinforcement learning of stimulus-response associations with a context-network that encodes the current task (called task-set network). To generalize across the three different possible stimuli in a given task the context network uses a Hebbian unsupervised learning rule. The most problematic part, from a biological perspective, is the fact that the feedback from the context network to the stimulus-response association network acts directly on the connections inside the stimulus-response association network (as opposed to input to neurons which would be the standard way of doing it). Unless this is justified by additional information (could be text, links to biological literature of biological synapses, or simulations of a more complex, but neuron-based model) I consider this as a potential flaw of the model. The link to the Fusi paper is not sufficient in my opinion.

We thank the reviewer for pointing this out. We fully agree that the feedback from the task-set network to the association network is biologically the most problematic part, and in fact the least constrained by known physiology, so that a number of mechanisms are possible. However, our main goal was to examine the hypothesis that chunking of stimulus-response associations provides an effective mechanism for task-set learning. From that perspective, we considered the details of the implementation of the feedback to be secondary, and this was our rationale for using a highly simplified implementation. Other implementations are the topic of ongoing work (see e.g. Dubreuil et al., 2019).

To address the reviewer’s comment, we have rewritten the Introduction to emphasise the fact that we are using an abstracted model (rather than a biologically plausible one) to test the hypothesis we are examining, and be able to fit the model to behavioral data (see also the response to Comment 10 below).

We have also rewritten the subsection “Biological plausibility of the model”. We now say:

“Despite the basic biological plausibility of the underlying mechanisms, implementing task-set learning in a more detailed model with heterogeneous connectivity and continuous-valued, or spiking neurons driving ongoing plasticity would remain a challenge (see e.g. Zenke et al., 2015). […] Ultimately, a biologically plausible instantiation of this gating remains an open question, as the currently available physiological data does not provide sufficient constraints.”

10) Motivation of model. Even though the authors imply that they have put together a biologically plausible model, it actually looks more like a very high-level abstract model where a whole population of neurons is replaced by a single binary switch; where there are exactly as many binary switches as there are potential combinations of stimuli and responses; and where learning rules are written as algorithmic updates rather than as weight changes driven by pre-and postsynaptic activity.Thus, I was not convinced about the approach until I understood in Figure 6 and the associated text that the authors actually use this abstract model (which only has a few free parameters, number to be clarified, see below) to fit behavioral data on a subject-by-subject basis. For me this is the strongest point of the paper which eventually makes me support publication after some modifications.Nevertheless, it is unclear to me whether the same learning rules would also work if the neural network model included a large number of randomly connected neurons, heterogeneity, and continuous-values rate units. I understand that the competitive dynamics would make the continuous-values units near-binary after learning, but the dynamics during learning might still be different, and hence convergence times as well (which could influence the results of the paper). Alternatively, reformulate the text to make it clear that the modeling work uses a rather abstract model with abstract learning rules.

We thank the reviewer for their comments. We fully agree that the strongest point of our paper is that we have a model with a small number of parameters that can be easily fit to subject-by-subject behavioral data. Although the model is biologically-motivated, it is far from being fully biologically plausible. Following the reviewer’s suggestion, in the revised version, we have strongly toned down the biological aspect, and, as suggested, clarify that we use an abstracted model to fit behavioral data, produce predictions, and correlate with BOLD data. To this end, we have reshaped the Introduction, and in particular removed the first paragraph on synaptic plasticity. In the Discussion section, we have rewritten the subsection “Biological plausibility of the model”. Throughout the text, we point out that we use an abstracted model that allows us to fit behavioral data on a subject by subject basis, in particular on:

– Introduction: “Here we use an abstracted network model to examine the hypothesis that task-sets are learnt through simple Hebbian plasticity based on temporal contiguity between different stimulus-response associations.”

– Subsection “A network model for learning task-sets by chunking stimulus-response pairs”: “we studied an abstracted neural network model (Figure 1D), that built on previous modeling studies of a trace conditioning task in monkeys (Fusi et al., 2007; Rigotti et al., 2010b). While the model included some basic biological constraints, it was purposefully simplified to allow for straightforward fitting to human behavioral data on a subject-by-subject basis.”

– Discussion section: “To allow for a direct comparison with human behavior data, we used a highly simplified network model with a small number of free parameters.”